# TabCBM: Concept-based Interpretable Neural Networks for Tabular Data

**Mateo Espinosa Zarlenga**                                    *me466@cam.ac.uk*
*Department of Computer Science and Technology*
*University of Cambridge*

**Zohreh Shams**                                              *zs315@cam.ac.uk*
*Department of Computer Science and Technology*
*University of Cambridge*

**Michael Edward Nelson**                            *mikeednelson92@gmail.com*
*Keyrock*
*European Bioinformatics Institute*
*University of Cambridge*

**Been Kim**[*]                                            *beenkim@google.com*
*Google DeepMind*

**Mateja Jamnik**[*]                                  *mateja.jamnik@cl.cam.ac.uk*
*Department of Computer Science and Technology*
*University of Cambridge*

**Reviewed on OpenReview:** *https://openreview.net/forum?id=TIsrnWpjQO*

## Abstract

Concept-based interpretability addresses the opacity of deep neural networks by constructing an explanation for a model's prediction using high-level units of information referred to as concepts. Research in this area, however, has been mainly focused on image and graph-structured data, leaving high-stakes tasks whose data is tabular out of reach of existing methods. In this paper, we address this gap by introducing the first definition of what a high-level concept may entail in tabular data. We use this definition to propose Tabular Concept Bottleneck Models (TabCBMs), a family of interpretable self-explaining neural architectures capable of learning high-level concept explanations for tabular tasks. As our method produces concept-based explanations both when partial concept supervision or no concept supervision is available at training time, it is adaptable to settings where concept annotations are missing. We evaluate our method in both synthetic and real-world tabular tasks and show that TabCBM outperforms or performs competitively compared to state-of-the-art methods, while providing a high level of interpretability as measured by its ability to discover known high-level concepts. Finally, we show that TabCBM can discover important high-level concepts in synthetic datasets inspired by critical tabular tasks (e.g., single-cell RNAseq) and allows for human-in-the-loop concept interventions in which an expert can identify and correct mispredicted concepts to boost the model's performance.

## 1 Introduction

Artificial Intelligence, spearheaded by advances in Deep Neural Networks (DNNs), has recently mastered tasks once believed to be solely achievable by virtue of innate human ingenuity (Krizhevsky et al., 2012; Jumper et al., 2021; Brown et al., 2020; Silver et al., 2017). Although impressive, these achievements have a

caveat: even though DNNs offer a powerful learning framework, their complexity obscures their reasoning, which hampers their applicability in tasks that require human-understandable explanations for predictions.

Concerns on the legal (Erdélyi & Goldsmith, 2018) and ethical (Durán & Jongsma, 2021; Lo Piano, 2020) ramifications of deploying such "black-box" models in real-world tasks have given rise to *explainable artificial intelligence* (XAI), methods that are paving the way for safe DNN deployment. Amongst these, *concept-based interpretability* (Kim et al., 2018; Koh et al., 2020a; Espinosa Zarlenga et al., 2022), where a DNN is explained via high-level human-understandable concepts, has recently gained attention. By constructing explanations that use low-dimensional human-understandable representations rather than input features, these methods reduce the mental load required to decode an explanation, thus enabling easy model inspection.

Given the novelty of concept-based explanations, their development has been limited in breadth, with only a few data modalities explored. As such, most concept-learning methods have focused on image (Ghorbani et al., 2019b; Chen et al., 2020b), sequential (Yeh et al., 2020; Kazhdan et al., 2020a), and graph-structured (Magister et al., 2021; 2022) data, leaving other modalities under-explored. Hence, crucial *tabular* tasks (e.g., genomics, clinical, and financial tasks), where DNNs have recently been successfully deployed (Kadra et al., 2021; Borisov et al., 2021), have been overlooked by the concept-based XAI literature. More importantly, while the definition of a concept in highly-structured modalities such as images (e.g., image segments (Ghorbani et al., 2019b)) or graphs (e.g., motifs (Magister et al., 2021)) is well-understood, tabular data does not have the spatial or geometric structure observed in these domains. This renders existing concept-based methods ill-posed for tabular tasks.

In this paper, we address this gap by defining a concept in tabular domains and proposing *TabCBM*, an interpretable neural architecture capable of learning tabular concept-based explanations both in the presence and absence of concept annotations. More broadly, our contributions are: (1) We propose, to the best of our knowledge, the first formalisation of a concept in tabular data; (2) we introduce TabCBM, an end-to-end concept-interpretable neural architecture that learns concept-based explanations for tabular tasks, both when presented with examples of such concepts (*concept-supervised*) or when no concept annotations are available (*concept-unsupervised*); (3) we show that TabCBM learns meaningful and interpretable concepts in complex datasets, such as single-cell RNAseq tasks, without sacrificing performance; and (4) we validate our method by demonstrating that it can be used in a human-in-the-loop setup where an expert may easily identify and intervene on its learnt concepts to improve model performance.

## 2 Background & Related Work

**Concept Learning**   Initial Concept Learning methods (Bau et al., 2017; Fong & Vedaldi, 2018) explored whether DNNs encode known high-level concepts within their learnt latent space. Aiming to circumvent known failure models (Kindermans et al., 2017; Ghorbani et al., 2019a) in popular saliency methods (Selvaraju et al., 2017; Sundararajan et al., 2017), as well circumvent the high mental load required to decode feature-level explanations, XAI algorithms such as TCAV (Kim et al., 2018), CaCE (Goyal et al., 2019), and CME (Kazhdan et al., 2020b) explored explaining a DNN using high-level concepts extracted from its latent space. Follow-up work (Koh et al., 2020a; Chen et al., 2020b), however, argued that extracting concept explanations after training may fail to capture conceptual relations with high fidelity. This has given rise to models that are "concept-aware" and instead construct concept-level explanations at inference.

*Concept Bottleneck Models* (CBMs) (Koh et al., 2020a) provide a framework to study most concept-based methods. Formally, a CBM is a composition of two models $g_\Phi$ and $f_\Theta$. The first model $g_\Phi : \mathcal{X} \to \mathcal{C}$, a "*concept encoder*" with parameters $\Phi$, maps input features $\mathbf{x} \in \mathcal{X}$ (e.g., a bird's image) to a $k$-dimensional concept representation $\hat{\mathbf{c}} \in \mathcal{C} \subseteq \mathbb{R}^k$ ($k$ is a property of the task). Each learnt concept's value $\hat{c}_i$ is incentivised to be aligned with a known ground-truth concept $c_i$ (e.g., "has white wings"). The second model $f_\Theta : \mathcal{C} \to \mathcal{Y} \subseteq [0,1]^L$, a "*label predictor*" with parameters $\Theta$, maps $\hat{\mathbf{c}} \in \mathcal{C}$ to a distribution over $L$ *downstream task* labels $y \in \mathcal{Y}$ (e.g., bird's type). A CBM can be trained, either *jointly*, *sequentially*, or *independently*, by learning $f_\Theta$ and $g_\Phi$ in a *concept-supervised* setting. In this setup, we are given a training set $\mathcal{D} := \big\{ (\mathbf{x}, \mathbf{c}, y) \mid \mathbf{x} \in \mathcal{X}, \mathbf{c} \in \{0,1\}^k, y \in \{1, \cdots, L\} \big\}$ where each sample $\mathbf{x}$ is annotated with a binary concept vector $\mathbf{c}$ and a task label $y$. Aligning $\hat{\mathbf{c}}$ to $\mathbf{c}$ using such a dataset encourages a CBM's "bottleneck" $\hat{\mathbf{c}} = g_\Phi(\mathbf{x})$ to serve as an

explanation for its prediction $\hat{y} = f_\Theta(g_\Phi(\mathbf{x}))$ and makes the CBM receptive to *concept interventions* (Koh et al., 2020a), where experts can improve a CBM's performance by correcting mispredicted concepts.

To circumvent the need for concept annotations during training, *concept-unsupervised* methods discover concepts using feedback from a downstream task. Post-hoc concept-unsupervised methods such as Concept Completeness-aware Discovery (CCD) (Yeh et al., 2020) learn a concept basis that contains enough information to reconstruct a DNN's latent space. Others, like ACE (Ghorbani et al., 2019b) and GCExplainer (Magister et al., 2021), instead discover concepts by clustering a DNN's latent space. In parallel, concept-unsupervised interpretable models, such as Self-explaining Neural Networks (SENNs) (Alvarez-Melis & Jaakkola, 2018), have also been proposed. Nevertheless, all of these methods have historically been applied to vision, graph, and language datasets, leaving their behaviour in tabular datasets as an open question.

## 3   What is a concept in a tabular task?

While it is clear what a concept is in images (e.g., image segments (Ghorbani et al., 2019b)), sequences (e.g., sentiment (Kazhdan et al., 2020a)) and graphs (e.g., subgraph motifs (Magister et al., 2021)), tabular data do not have the well-defined structure observed in these other domains. Nevertheless, it has been observed that, in tabular datasets, one commonly finds high inter-correlations or multicollinearity amongst features (Belsley et al., 2005; Wang et al., 2015; Lee et al., 2021). For example, in genomics, where a sample is formed by thousands of gene expressions, major diseases can be mostly attributed to the interaction of a few subsets of related genes (Jackson et al., 2018; Lee et al., 2021). Similarly, in single-cell RNAseq (sc-RNAseq) tasks (Macaulay & Voet, 2014) a cell's state can be decomposed into groups of highly related genes, known as Gene Expression Programs (GEPs) (Segal et al., 2003; Kotliar et al., 2019), and groups of transcription factors whose expression or lack thereof represent simple biological functions (e.g., undergoing cell division).

**Definition**   Inspired by the feature multicollinearity observed in tabular tasks, we propose that fixed subsets of highly correlated features can be thought of as tabular concepts. Specifically, we argue that a subset of correlated features defines a concept if its features are the inputs of a possibly nonlinear scoring function $s^{(i)}$ whose output represents a high-level "meta-feature"'s activation (e.g., a GEP's activation). We formalise this by defining a tabular concept on an $n$-dimensional task as a tuple $(\pi^{(i)}, s^{(i)})$ where $\pi^{(i)} \in \{0,1\}^n$ is a mask indicating the input features that are related to this concept and $s^{(i)} : \mathbb{R}^{\sum_j \pi_j^{(i)}} \to \{0,1\}$ is a function indicating the concept's activation given the features selected by $\pi^{(i)}$ (i.e., features $j$ with $\pi_j^{(i)} = 1$).

To see how this definition can be applied to a real-world example, consider the task of predicting a user's credit score from their past financial history. Previous work (Chen et al., 2018a) has shown that grouping a potential client's past "criminal" loan history (e.g., "number of defaulted loans", "longest delayed payment", "last delayed payment", etc.) into a single "delinquency" score can help predict the likelihood of default. In this context, scoring models, such as that proposed by Chen et al. (2018a), can use a simple interpretable model $h$ that predicts a "delinquency" score from known delinquency-related features to train a simple downstream model that predicts the likelihood of loan default. Hence, the subset of features needed for $h$ can be thought of as a high-level concept when predicting a loan's risk. We can then formally describe this as a tabular concept by letting $\pi^{(i)}$, the mask selecting all the features needed to compute this concept, be a binary vector indicating each delinquency-related feature and $s^{(i)}$, the concept's scoring function, be $h$.

## 4   Tabular Concept Bottleneck Model

**Motivation**   A key component of our tabular concept definition is that of understanding which subsets of features form concepts. Here, we explore discovering these subsets by building upon a related field: feature selection (Liu et al., 1996; Kohavi & John, 1997). Methods in this field reduce the noise in high-dimensional data by selecting a small subset of relevant features to learn from (Chandrashekar & Sahin, 2014). Recent advancements in deep learning (Yamada et al., 2020; Huang et al., 2020b), and in particular in self-supervised learning (Ucar et al., 2021; Yoon et al., 2020; Lee et al., 2021), introduced differentiable neural architectures for feature selection. In our work, we build on Self-supervision Enhanced Feature Selection (SEFS) (Lee et al., 2021) to design our concept-based architecture. As we discuss below, the crux of our approach lies in

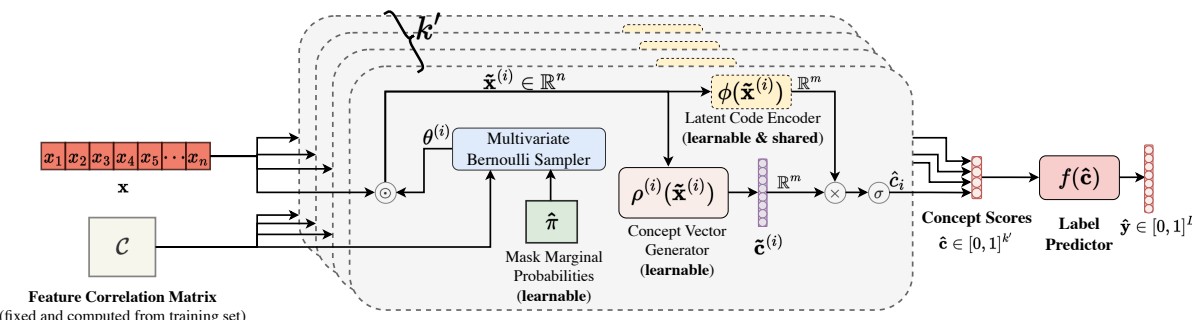

Figure 1: **TabCBM**: for each learnt concept (here shown for $k' = 4$) we learn both a per-concept feature importance mask $\hat{\pi}^{(i)}$ and a concept score $\hat{c}_i \in [0, 1]$ that indicates the concept's activation. All concatenated scores are passed to a differentiable label predictor to output a downstream prediction.

framing tabular concept discovery as learning multiple feature selection masks simultaneously from which multiple concept activation scores can be derived and used for downstream tasks.

**Model Architecture**  A *Tabular Concept Bottleneck Model* (TabCBM), shown in Figure 1, takes as an input an $n$-dimensional tabular sample $\mathbf{x} \in \mathbb{R}^n$ and discovers $k'$ tabular concepts, where $k'$ is a user-selected hyperparameter. Inspired by our definition of a tabular concept, TabCBM aims to discover a set of $k'$ tabular concepts $\{(\hat{\pi}^{(i)}, s^{(i)})\}_{i=1}^{k'}$ from which a downstream task label $\hat{y}$ can be predicted, and explained, via a concept score vector $\hat{\mathbf{c}} = [s^{(1)}(\hat{\pi}^{(1)} \odot \mathbf{x}), \cdots, s^{(k')}(\hat{\pi}^{(k')} \odot \mathbf{x})]^T$. We proceed by relaxing our discovery task so that each concept's mask $\hat{\pi}^{(i)} \in [0, 1]^n$ and scoring function $s^{(i)} : \mathbb{R}^n \to [0, 1]$ operate within a continuous space $[0, 1]$. Such relaxation allows concept discovery in a differentiable manner.

Given a training set $\mathcal{D} := \{(\mathbf{x}_l, y_l)\}_{l=1}^N$ with $N$ samples and labels, we train TabCBM as follows: for each concept $i \in \{1, \cdots, k'\}$, we learn a "soft" feature importance mask $\hat{\pi}^{(i)} \in [0, 1]^n$ that indicates how "important" each input feature is for concept $i$. Each mask can be thought of as a feature-selection subproblem and is learnt by adapting the feature selection process used in SEFS. Specifically, the vector $\hat{\pi}^{(i)}$ is used to indicate *the marginal probabilities* of a differentiable *multivariate Bernoulli sampler* which, by looking at the empirical feature correlation matrix $\mathcal{C}$, samples a per-sample masking vector $\theta^{(i)} \in \{0, 1\}^n$ that *preserves empirical inter-feature dependencies*[1]. This means that, for example, in a genomics task where two genes are highly correlated, the sampler will be more likely to mask these two features together than masking one but not the other. This prevents information from masked features from leaking via unmasked but highly-related features. In practice, we use the Gaussian copula (Nelsen, 2007) described by Lee et al. (2021) and the Bernoulli relaxation and reparameterisation trick described by Wang & Yin (2020) for the differentiable multivariate Bernoulli sampler (see Appendix B for details). This means that $\theta^{(i)}$ will be continuous.

Once $\theta^{(i)}$ is generated, we mask the input features to compute a new sample whose unselected inputs are near zero $\tilde{\mathbf{x}}^{(i)} := \mathbf{x} \odot \theta^{(i)}$. When constructing our TabCBM model, we associate each concept with a learnable model $\rho^{(i)} : \mathbb{R}^n \to \mathbb{R}^m$ that takes $\tilde{\mathbf{x}}^{(i)}$ and generates a concept embedding representation $\rho^{(i)}(\tilde{\mathbf{x}}^{(i)})$ for concept $i$. Intuitively, $\rho^{(i)}(\tilde{\mathbf{x}}^{(i)})$ can be thought of as a per-sample embedding representation for the $i$-th concept and can be used to generate a concept score indicating whether this concept is "activated" in $\mathbf{x}$. Using a learnable model $\phi : \mathbb{R}^n \to \mathbb{R}^m$ that generates an $m$-dimensional latent representation from $\tilde{\mathbf{x}}^{(i)}$ (shared across *all* concepts and learnt end-to-end with TabCBM), we consider the inner product between a concept embedding and $\phi(\tilde{\mathbf{x}}^{(i)})$ to be a measure of concept "activation". We generate a scalar *concept activation score* $\hat{c}_i \in [0, 1]$ through the sigmoidal inner product:

$$s^{(i)}(\mathbf{x}) := \hat{c}_i = \sigma\big(\phi(\tilde{\mathbf{x}}^{(i)})^T \rho^{(i)}(\tilde{\mathbf{x}}^{(i)})\big). \tag{1}$$

Although future work may explore different scoring functions, here we opt to use the sigmoidal inner product between $\phi(\tilde{\mathbf{x}}^{(i)})$ and $\rho^{(i)}(\tilde{\mathbf{x}}^{(i)})$ rather than, say, the cosine similarity between these two vectors. Our choice of

---

[1]This is only done at training time to incentivise concept discovery. During testing, we deterministically set $\theta^{(i)} = \hat{\pi}^{(i)}$.

activation allows our model to utilise the magnitude of these vectors to help it more easily express a concept's activation or deactivation when passed to a sigmoidal function. This avoids the need to threshold the inner product to generate a concept score between 0 and 1, as done by CCD, and leads to better downstream performance (see Appendix J). Furthermore, notice that TabCBM generates its latent representation and concept scores using the masked sample $\tilde{\mathbf{x}}^{(i)}$, therefore avoiding information from irrelevant features leaking into a concept's score. Once all concept scores have been computed, they are concatenated into a concept bottleneck $\hat{\mathbf{c}} \in [0,1]^{k'}$ which is passed to a label predictor model $f : [0,1]^{k'} \rightarrow [0,1]^L$ to predict the downstream task. Once all concept scores have been computed, they can be concatenated to form a concept bottleneck representation $\hat{\mathbf{c}} \in [0,1]^{k'}$ which is passed to a label predictor model $f : [0,1]^{k'} \rightarrow [0,1]^L$ to predict the downstream task.

**Desiderata**  We argue that meaningful concept generators $\{\rho^{(i)}\}_{i=1}^{k'}$ and masks $\{\hat{\pi}^{(i)}\}_{i=1}^{k'}$, must satisfy:

1. **Concept Completeness**: one should be able to predict the labels $y$ of a specific downstream task of interest from the concept scores $\hat{\mathbf{c}}$.
2. **Diversity**: concept embeddings $\{\rho^{(i)}(\tilde{\mathbf{x}}^{(i)})\}_{i=1}^{k'}$ should each capture semantically unique concepts.
3. **Coherence**: If two samples are very similar, under some similarity metric (e.g., Euclidean distance), their concept scores $\hat{\mathbf{c}}$ should be equally as similar. This should hold even when determining similarity in a latent representation space (e.g., representations learnt by a feature extractor such as PCA).
4. **Specificity**: learnt concepts should be a function of only a small subset of input features (i.e., feature importance masks $\{\hat{\pi}^{(i)}\}_{i=1}^{k'}$ should be sparse).

We incorporate these desiderata into our model's objective function using a weighted loss of four components:

$$\mathcal{L}_{\text{concept-unsup}}(\mathbf{x}, y) := \mathcal{L}_{\text{task}}(f(\hat{\mathbf{c}}), y) + \lambda_{\text{co}}\mathcal{L}_{\text{co}}(\mathbf{x}) + \lambda_{\text{div}}\mathcal{L}_{\text{div}}(\mathbf{x}) + \lambda_{\text{spec}}\mathcal{L}_{\text{spec}}(\hat{\pi}^{(1)}, \cdots, \hat{\pi}^{(k')}) \qquad (2)$$

where $\lambda_{\text{co}}$, $\lambda_{\text{div}}$, and $\lambda_{\text{spec}}$ control how much we weigh each property.

The form taken by the task loss $\mathcal{L}_{\text{task}}$is task-specific (e.g., cross-entropy) and incentivises concept completeness. As in previous works (Alvarez-Melis & Jaakkola, 2018; Yeh et al., 2020), this loss function aims to discover concepts that jointly encode all the information necessary for predicting the downstream task. In light of the impossibility of discovering disentangled latent representations for a given distribution without any supervision (Locatello et al., 2019), this loss, therefore, provides the necessary feedback to discover important concepts for this task in the absence of full concept supervision.

For both the coherence and diversity loss terms, we extend the regularisers proposed by Yeh et al. (2020) by letting them operate on per-sample concept vectors. Specifically, our coherence loss incentivises the model to generate similar concept scores for samples whose latent representations are similar (according to their $\ell_2$ distance). This is achieved by minimising:

$$\mathcal{L}_{\text{co}}(\mathbf{x}_1, \cdots, \mathbf{x}_N) := -\frac{1}{Nt} \sum_{\mathbf{x}_i \in \{\mathbf{x}_1, \cdots, \mathbf{x}_N\}} \sum_{\phi(\mathbf{x}_j) \in \Psi_t(\phi(\mathbf{x}_i))} \frac{\hat{\mathbf{c}}(\mathbf{x}_i)^T \hat{\mathbf{c}}(\mathbf{x}_j)}{||\hat{\mathbf{c}}(\mathbf{x}_i)|| \, ||\hat{\mathbf{c}}(\mathbf{x}_j)||} \qquad (3)$$

where $\hat{\mathbf{c}}(\mathbf{x}_i)$ is the concept score vector for $\mathbf{x}_i$ and $\Psi_t(\phi(\mathbf{x}_i))$ represents the set of $t$-nearest-neighbours of $\phi(\mathbf{x}_i)$, selected using their $\ell_2$ distances, in the training set $\{\phi(\mathbf{x}_l)\}_{l=1}^N$ ($t$ is a hyperparameter selected as suggested by Yeh et al. (2020)). Similarly, we define the concept diversity loss as:

$$\mathcal{L}_{\text{div}}(\mathbf{x}_1, \cdots, \mathbf{x}_N) := \frac{1}{Nk'(k'-1)} \sum_{\mathbf{x} \in \{\mathbf{x}_1, \cdots, \mathbf{x}_N\}} \sum_{i=1}^{k'} \sum_{\substack{j=1 \\ j \neq i}}^{k'} \frac{\rho_j(\tilde{\mathbf{x}}^{(j)})^T \rho_i(\tilde{\mathbf{x}}^{(i)})}{||\rho_j(\tilde{\mathbf{x}}^{(j)})|| \, ||\rho_i(\tilde{\mathbf{x}}^{(i)})||}. \qquad (4)$$

That is, we learn diverse concepts by minimising the cosine similarity between distinct concept vectors for a sample. This loss term is similar to that used in several state-of-the-art contrastive learning pipelines (Chen et al., 2020a; Zbontar et al., 2021). Finally, we incentivise specificity by minimising the $\ell_1$ norm of all feature importance masks:

$$\mathcal{L}_{\text{spec}}(\hat{\pi}^{(1)}, \cdots, \hat{\pi}^{(k')}) := \frac{1}{k'n} \sum_{i=1}^{k'} ||\hat{\pi}^{(i)}||_1 \qquad (5)$$

This loss, as in several Lasso-based sparse classifiers (Tibshirani, 1996; Liu et al., 2017; Margeloiu et al., 2022), increases concept interpretability by forcing the selection of only a handful of features per concept.

**Providing partial concept supervision** TabCBM can benefit from available training concept supervision, even if given only for some training samples. Assume that a subset $X_{\text{concept-annotated}} \subseteq \mathcal{D}$ of our training data has $k$ binary ground-truth concept annotations $C := \left\{ \mathbf{c}_i \in \{0,1\}^k \mid (\mathbf{x}_i, y_i) \in X_{\text{concept-annotated}} \right\}$. This gives us a dataset of triplets $\mathcal{D}_{\text{concept-sup}} := \{(\mathbf{x}_i, \mathbf{c}_i, y_i)\}_i$. During training, assuming $k' \geq k$, we can then encourage specific concept scores to be aligned with ground-truth concepts in $C$ by minimising

$$\mathcal{L}_{\text{total}} = \mathbb{E}_{(\mathbf{x},y)\sim\mathcal{D}}\big[\mathcal{L}_{\text{concept-unsup}}(\mathbf{x}, y)\big] + \lambda_{\text{concept-sup}}\mathbb{E}_{(\mathbf{x},\mathbf{c},y)\sim\mathcal{D}_{\text{concept-sup}}}\big[\mathcal{L}_{\text{concept-sup}}(\mathbf{x}, \mathbf{c})\big] \tag{6}$$

where $\lambda_{\text{concept-sup}}$ controls how much we value *concept alignment* with the training concepts vs *concept discovery*, and $\mathcal{L}_{\text{concept-sup}}(\mathbf{x}, \mathbf{c})$ is the mean cross-entropy loss between $\hat{c}_i$ and $c_i$:

$$\mathcal{L}_{\text{concept-sup}}(\mathbf{x}, \mathbf{c}) := -\frac{1}{k}\sum_{i=1}^{k}\big(c_i \log(\hat{c}_i) + (1 - c_i)\log(1 - \hat{c}_i)\big). \tag{7}$$

This loss encourages the first $k$ concept scores to be aligned with known ground truth concepts $\mathbf{c}$ while incentivising the rest of concept scores to discover semantically distinct concepts to those given during training. This enables TabCBM to maintain high performance without many concept annotations while reducing concept impurities (Espinosa Zarlenga et al., 2023) and leakage (Mahinpei et al., 2021) in its bottleneck.

## 5 Experiments

In this section, we evaluate TabCBM by focusing on four sets of research questions:

- **Performance (Q1)** — Is TabCBM's performance competitive against that of "black-box" models and other concept-based models when trained with different degrees of concept supervision?
- **Concept Score Alignment (Q2a)** — Do concept scores predicted by TabCBM align with ground truth tabular concepts in the input task?
- **Concept Mask Alignment (Q2b)** — Even when the concept scores are aligned with known ground-truth concepts, this does not guarantee that the learnt feature importance masks match their corresponding ground-truth concept feature masks. Is it the case that TabCBM's concept masks capture subsets of features known to have some high-level interpretation? If so, can we use these masks to provide global feature importance?
- **Human-in-the-loop Interventions (Q3)** — Can we improve the downstream task performance of TabCBM if an expert can identify the semantics of a specific concept and intervene on it?

To answer these questions, we first evaluate TabCBM's task and concept accuracy in a variety of concept-supervised and concept-unsupervised setups. Next, we quantitatively and qualitatively study the concepts discovered by TabCBM when no concept supervision is given. Finally, we explore how TabCBM's performance can be boosted in a human-in-the-loop setting with the introduction of test-time concept interventions.

**Datasets** We evaluate our method on both synthetic and real-world tabular datasets. We construct four synthetic tabular datasets of increasing complexity: **Synth-Linear**, **Synth-Nonlin**, **Synth-Nonlin-Large**, and **Synth-scRNA**. In these datasets, each sample $\mathbf{x}$ is annotated with a downstream categorical task label $y$ and a set of $k$ ground truth tabular concepts $\{(\pi^{(i)}, s^{(i)})\}_{i=1}^{k}$ ($k = 5$ for *Synth-Nonlin-Large*, $k = 11$ for *Synth-scRNA*, and $k = 2$ otherwise). By construction, $s^{(i)}(\mathbf{x})$ depends only on a small number of input features for all $i$ and the label $y$ is uniquely described by the sample's concept activation vector $\mathbf{c} := [s^{(1)}(\mathbf{x}), \cdots, s^{(k)}(\mathbf{x})]^T$. **Synth-scRNA**, in particular, is a synthetic single-cell RNA high-dimensional task in which each concept represents one of 11 known GEPs, 10 of which are *identity GEPs* (i.e., they are aligned with the sample's cell type) and one which is an *activity GEPs* (i.e., a group of co-regulated genes and transcription factors which represent environmental/epigenetic biological functions). This makes this task representative of an impactful application of TabCBM due to the importance of discovering GEPs from

Table 1: Downstream accuracy (%) and standard errors show that TabCBM (trained without concept supervision) outperforms or is on par with all methods across tasks. Values are rounded up to two decimals and errors are not shown if they are less than $10^{-5}$. Because CCD is a post-hoc method, its reported accuracy corresponds to that obtained when reconstructing its DNN's latent space with its concept scores.

| Dataset | TabCBM (ours) | SENN | CCD (recon) | MLP | TabNet | TabTransformer | XGBoost | LightGBM |
|---|---|---|---|---|---|---|---|---|
| Synth-Linear | **99.84 ± 0.06** | 98.15 ± 0.2 | 96.47 ± 1.3 | 97.57 ± 0.37 | 97.57 ± 0.37 | 82.91 ± 0.55 | 96.43 | 96.8 |
| Synth-Nonlin | **98.36 ± 0.15** | 89.14 ± 0.71 | 85.99 ± 2.28 | 87.65 ± 0.98 | 91.57 ± 0.48 | 81.07 ± 0.83 | 88.43 | 89.33 |
| Synth-Nonlin-Large | **62.78 ± 1.13** | 49.78 ± 2.08 | 51.64 ± 1.71 | 40.73 ± 6.42 | 51.01 ± 2.57 | 54.63 ± 1.17 | 22.48 ± 0.48 | 23.58 ± 0.78 |
| Synth-scRNA | **93.66 ± 1.41** | 78.32 ± 3.03 | 68.83 ± 1.73 | 73.87 ± 1.43 | 90.66 ± 1.10 | 87.29 ± 0.68 | 90.44 ± 1.06 | 89.96 ± 1.57 |
| Higgs (without high-level) | **80.42 ± 0.3** | 70.61 ± 0.12 | 77.84 ± 0.08 | 79.90 ± 0.15 | 79.44 ± 0.16 | 74.94 ± 0.21 | 68.85 ± 0.02 | 68.87 ± 0.06 |
| Higgs (with high-level) | **78.62 ± 0.12** | 73.53 ± 0.71 | 77.92 ± 0.09 | 78.44 ± 0.02 | 78.12 ± 0.05 | 74.22 ± 0.42 | 75.33 ± 0.04 | 75.33 ± 0.03 |
| PBMC | **93.35 ± 0.16** | 92.24 ± 0.23 | 93.14 ± 0.30 | 91.66 ± 1.95 | 92.74 ± 0.46 | 91.01 ± 0.33 | 93.09 ± 0.29 | 93.01 ± 0.24 |
| FICO | 72.08 ± 0.42 | 66.78 ± 0.69 | 65.46 ± 4.91 | 67.98 ± 1.36 | 71.20 ± 0.87 | 65.66 ± 0.85 | 72.33 ± 0.44 | **72.63 ± 0.12** |

sc-RNAseq data in genomics (Kotliar et al., 2019; Kharchenko et al., 2014; Satija et al., 2015). All of these synthetic datasets allow TabCBM to be evaluated in a controlled environment where we know the underlying set of complete tabular concepts and their relationship to the downstream task.

Finally, we use three real-world datasets with unknown ground-truth concepts: (1) **PBMC** (10x Genomics, 2016a;b) as a high-dimensional single-cell transcriptomic dataset, (2) **Higgs** (Aad et al., 2012) as a large real-world physics tabular dataset (we also include a variant of this dataset without the hand-crafted high-level features defined by Baldi et al. (2014)), and (3) **FICO** (Fair Isaac Corporation, 2019) as a high-stakes financial task whose features are both categorical and continuous. For further details of each dataset used in our evaluation, refer to Appendix D.

**Baselines and model selection** We compare TabCBM against other concept-based methods such as CBMs and Concept Embedding Models (CEMs) (Espinosa Zarlenga et al., 2022) when concept supervision is provided, and against SENN and CCD when concept annotations are not provided. Similarly, we evaluate TabNet (Arık & Pfister, 2021), TabTransformer (Huang et al., 2020a), Multi-layer Perceptrons (MLPs), XGBoost (Chen & Guestrin, 2016) and LightGBM (Ke et al., 2017) as representative methods of high-performing tabular models, where TabNet, TabTransformer and MLP are neural models and the others are gradient-boosted models (GBMs). When studying TabCBM's concept representations, we include Principle Component Analysis (PCA) (Pearson, 1901) as a representation-learning baseline due to its common use in high-dimensional tabular datasets. For the sake of fairness, when possible we train all models using the same architecture/capacity and training configuration. Similarly, we select a model's hyperparameters via a grid search over a set of possible values and report test scores for the hyperparameters that maximise the validation loss. In particular, unless specified otherwise, the number of concepts in concept-learning methods (i.e., $k'$) is set to the number of ground truth concepts $k$. If $k$ is unavailable for the task (e.g., as in FICO), we estimate $k'$ by selecting the value that maximises the validation loss over a fixed set of candidate values. Similarly, for both CCD and TabCBM the number of top-$t$ neighbours for their coherence loss is selected as suggested by Yeh et al. (2020). Furthermore, when training neural models we map categorical features to continuous scalars using a 1D embedding and, to speed up training, we pre-train TabCBM's latent encoder and mask generators using the process described by Lee et al. (2021) (see Appendix C for details). Finally, when presenting our results, we average each metric over five models trained with different initialisations. For further details on our training setup, see Appendix E.

## 5.1 Model Performance (Q1)

In this section, we evaluate TabCBM's first desideratum, *concept completeness*, by contrasting its downstream task accuracy against that of competing methods. The objective of these experiments is to explore whether our method can maintain a competitive performance against state-of-the-art baselines across several tasks. For this, we study two training setups: (1) a *concept unsupervised* setup (i.e., no concept annotations), and (2) a *partially* and *fully* concept supervised setup (i.e., a fraction of samples are concept-annotated).

**Concept unsupervised** Our experiments' results, shown in Table 1, suggest that TabCBM's task accuracy is competitive against that of black box and concept-interpretable models. Particularly, TabCBM

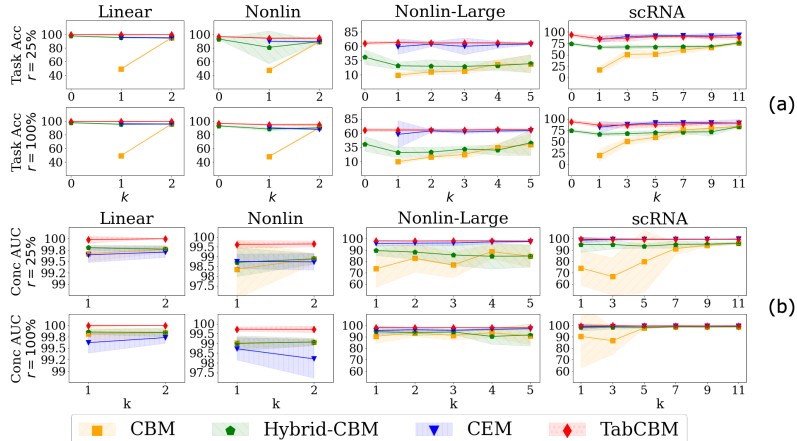

Figure 2: (a) Task accuracy (%) and (b) mean concept AUC (%) in synthetic tasks. We vary the number of supervised concepts $k$ (x-axis) and the fraction $r$ of concept-annotated samples (25% and 100% of samples for the top and bottom rows, respectively). We include only concept-supervised methods, disqualifying CCD and SENN. Similarly, only TabCBM and Hybrid-CBM have results shown when the number of training concepts is 0 (i.e., $x = 0$) as CBM and CEM do not support training without available concept labels.

significantly outperforms existing concept-based architectures (i.e., CCD and SENN) and neural models (i.e., MLP, TabNet, TabTransformer) across all tasks. Similarly, we observe that TabCBM outperforms XGBoost and LightGBM in most tasks with the exception of the FICO dataset, in which TabCBM is marginally outperformed by gradient-boosted methods potentially due to the discrete nature of this task's features and the small training set (something which has been previously observed for other neural models in financial tasks (Schmitt, 2022)). Nevertheless, our aggregate results strongly indicate that TabCBM does not sacrifice the performance seen in state-of-the-art black-box models when it is not provided with supervision.

**Concept supervised**  Next, we explore TabCBM's task and concept accuracy when it is provided with concept supervision. For this, we compare TabCBMs against equivalent-capacity CEMs, CBMs, and Hybrid-CBMs (Mahinpei et al., 2021; Espinosa Zarlenga et al., 2022), the latter being a CBM with extra unsupervised bottleneck activations (we add as many activations as TabCBM's bottleneck). We train all models on our synthetic datasets, given that they have ground-truth concept annotations, while we vary the number of supervised concepts. This is done to study TabCBM's sensitivity to concept incompleteness and is implemented by selecting a fixed random subset of concepts we provide annotations for. Finally, we evaluate each method's data efficiency w.r.t. the number of concept-annotated samples by providing concept supervision for 100% and 25% of the training samples (see Appendix F for details).

Our task results, shown in Figure 2a, suggest that TabCBM attains better task predictive performance than CEMs, CBMs, and Hybrid-CBMs. We observe this regardless of (i) the number of annotated concepts (x-axis) and (ii) the number of concept-annotated samples (TabCBM's task accuracy is nearly constant in both rows of Figure 2). Furthermore, we observe this improvement, which is significant in complex datasets, comes without sacrificing TabCBM's mean concept predictive ROC-AUC[2] for the concepts that receive supervision, as seen by TabCBM's competitive concept accuracy in Figure 2b. The same cannot be said of CBMs which struggle to find a balance between concept interpretability and task accuracy when the training set of concepts is incomplete (see Figure 2a when a small number of training concepts is provided).

Notice that although TabCBM's task accuracy is generally unaffected by the introduction of concept supervision (as opposed to Hybrid-CBM's), we observe an exception in Synth-scRNA. We see that TabCBM's performance in Synth-scRNA drops when concept supervisions are introduced, suggesting that an accuracy-interpretability trade-off may exist in complex tasks. Nevertheless, as discussed in Appendix K, when provided with enough concept supervision this trade-off causes TabCBM to be only marginally worse than

---

[2]This is defined as the mean area under the ROC when predicting each training concept $\mathbf{c}_i$ with concept score $\hat{c}_i$.

the next best-performing model. In practice, this may be alleviated by fine-tuning $\lambda_{\text{concept-sup}}$ and it may be a negligible sacrifice considering that it leads to high-fidelity concept explanations for TabCBM's predictions while allowing performance-boosting test-time concept interventions as we will see in Section 5.4.

## 5.2 Concept Score Alignment (Q2a)

In this section, we explore whether TabCBM learns concepts that align with known ground-truth concepts when it receives no concept supervision. We show that, under a series of reasonable quantitative metrics, TabCBM's concepts are better aligned with known ground-truth concepts than those learnt by other concept-based methods. This implies that our method, even without any concept supervision, is able to discover concepts that are semantically aligned to high-level units of information that experts may use in the same task. We follow this quantitative analysis by qualitatively studying TabCBM's concepts in *Synth-scRNA* and showing a clear alignment between TabCBM's concepts and known GEPs.

**Quantitative results** We investigate the alignment of TabCBM's concept explanations/scores by considering different metrics that capture TabCBM's desiderata. For this, we compare TabCBM's concept-based explanations against those learnt by CCD, SENN, and PCA by looking at (a) their predictive power w.r.t. ground-truth concepts, (b) their alignment to known ground-truth concepts, and (c) their diversity. We first capture the predictive power of learnt concept representations w.r.t. ground truth concepts using the DCI Informativeness score (*Info*) (Eastwood & Williams, 2018b), which measures the information content of discovered concepts w.r.t. ground truth concepts. Then, we capture an explanation's alignment with known ground-truth concepts by checking whether learnt concept scores lead to (i) coherent clusters w.r.t. known ground-truth concepts, as captured by a high Concept Alignment Score (CAS) (Espinosa Zarlenga et al., 2022), (ii) representations that have at least one aligned dimension per ground truth concept, as captured by a high DCI-Completeness (*Compl*) (Eastwood & Williams, 2018b), and (iii) representations that have one and only one dimension aligned to a ground-truth concept, as captured by a high $R^4$ score (Ross & Doshi-Velez, 2021). Finally, we quantify the diversity of the set of learnt concepts by computing their Mutual Information Gap (*MIG*) (Chen et al., 2018b) and Disentanglement Score (*Dis*) (Eastwood & Williams, 2018b), both measurements of whether each ground truth concept is aligned with one and only one learnt concept, complementing the $R^4$ with metrics that do not assume invertible correspondences. We provide a summary of these metrics in Appendix L.

Table 2: Mean concept representation quality metrics (%) measured across all synthetic datasets for concept-based methods (higher values are better). We show the desideratum closely captured by each metric in *italics* and parenthesis. To disambiguate variances across tasks, we also show each method's relative mean rank $\bar{r}$.

| | CAS (*coherence*) | MIG (*diversity*) | $R^4$ (*coherence & diversity*) | Dis (*diversity*) | Compl (*completeness*) | Info (*completeness*) |
|---|---|---|---|---|---|---|
| **TabCBM (ours)** | **87.55 ± 14.07** ($\bar{r}$ = 1.5) | **57.71 ± 26.27** ($\bar{r}$ = 1.5) | **78.36 ± 17.65** ($\bar{r}$ = 1.5) | **69.83 ± 23.65** ($\bar{r}$ = 1.5) | **70.44 ± 22.81** ($\bar{r}$ = 1.5) | **80.83 ± 11.22** ($\bar{r}$ = 1.25) |
| SENN | 60.11 ± 6.26 ($\bar{r}$ = 2.75) | 9.92 ± 5.68 ($\bar{r}$ = 3.5) | 30.83 ± 17.38 ($\bar{r}$ = 3.5) | 21.49 ± 6.51 ($\bar{r}$ = 3.5) | 29.56 ± 7.30 ($\bar{r}$ = 3.75) | 32.50 ± 25.82 ($\bar{r}$ = 3.25) |
| CCD | 52.86 ± 20.82 ($\bar{r}$ = 3) | 29.57 ± 5.86 ($\bar{r}$ = 2) | 65.79 ± 10.49 ($\bar{r}$ = 2) | 39.66 ± 5.89 ($\bar{r}$ = 2) | 41.04 ± 6.93 ($\bar{r}$ = 2.25) | 66.77 ± 9.30 ($\bar{r}$ = 2.5) |
| PCA | 57.54 ± 12.89 ($\bar{r}$ = 2.75) | 9.48 ± 5.73 ($\bar{r}$ = 3) | 19.59 ± 28.18 ($\bar{r}$ = 3) | 24.15 ± 16.9 ($\bar{r}$ = 3) | 36.17 ± 15.86 ($\bar{r}$ = 2.25) | 18.7 ± 32.68 ($\bar{r}$ = 3) |

We compute these metrics across all of our synthetic tasks, as they have concept annotations, and show their averages across all of these tasks in Table 2. Our results suggest that TabCBM significantly outperforms all other concept-learning methods across all metrics. Specifically, high CAS and $R^4$ scores for TabCBM suggest that it is learning concepts that are highly aligned to known ground truth concepts. Similarly, high *Dis* and *MIG* scores suggest that TabCBM's concepts are disentangled and diverse. Finally, as indicated by TabCBM's high *Info* and *Compl* scores, TabCBM effectively captures significant information needed for predicting the set of ground truth concepts, therefore learning complete explanations of its task and explaining the observed higher performance of TabCBM over other concept-based methods in Section 5.1.

**Qualitative Results** To qualitatively corroborate our previous results, we visualise TabCBM's discovered concepts in Synth-scRNA using UMAP (McInnes et al., 2018), a commonly-used dimensionality reduction method for scRNA data. For each ground truth concept $c_i$, we visualise the Synth-scRNA dataset (both train and test data) via its UMAP projection while colouring each cell's representation according to $c_i$. Then, we select the discovered TabCBM concept whose score $\hat{c}_j$ has the highest absolute Pearson correlation

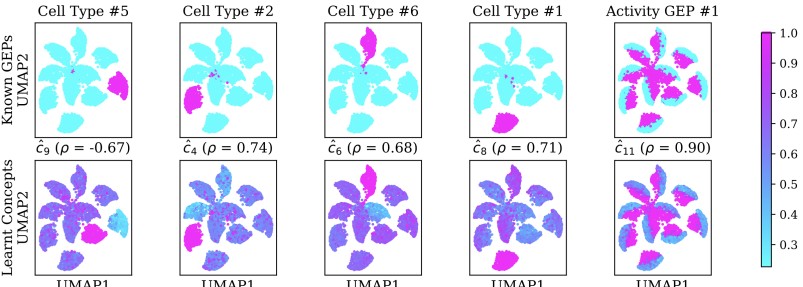

Figure 3: UMAP of Synth-scRNA showing five randomly selected ground truth GEPs (top row with the right-most plot showing the only activity GEP) and the score of TabCBM's learnt concept with the highest mutual information (captured by the *absolute* Pearson correlation $\rho$) with each GEP (bottom row). Note that because TabCBM may learn a ground-truth concept's complement labelling, we use the absolute correlation to capture strong negative (e.g., $\hat{c}_9$) correlations as well as strong positive correlations (e.g., $\hat{c}_{11}$). We highlight that the TabCBM model whose scores are shown in this plot *did not* receive any concept supervision.

coefficient with $c_i$ and recreate the plot by colouring each cell using $\hat{c}_j$. This allows us to visually inspect $c_i$'s alignment with a (linearly) strongly correlated learnt concept score in TabCBM. Notice that although other proxies for mutual information could be used (e.g., Gini impurity), we opt for Pearson correlation due to its efficiency. Similarly, because we only study a cell's cluster membership, our analysis avoids known issues with hypothesising from low-dimensional data visualisations (Wattenberg et al., 2016). Our results in Figure 3 show that TabCBM's concepts are closely aligned with ground truth concepts. In particular, we see that TabCBM discovers cell types/identity GEPs (e.g., $\hat{c}_4$ and $\hat{c}_9$) and the task's only activity GEP (i.e., $\hat{c}_{11}$). As seen in Appendix G, the same cannot be said of concepts discovered by CCD and SENN.

### 5.3 Concept Mask Alignment (Q2b)

An advantage of TabCBM over existing concept-based methods is that its concepts allow for easy identification of the input features that compose them. In this section, we study TabCBM's learnt concept masks to explore whether they can (1) identify features that are known to be important for ground-truth concepts, and (2) provide a sense of global feature importance through mask aggregation. Our experiments show that both properties hold and, therefore, experts analysing data with TabCBM may use its learnt concept masks to discover novel interactions between groups of input features that are important for their task of interest.

**Ground-truth concept mask alignment** First, we explore whether TabCBM's feature importance masks align with sets of features known to compose each ground truth concept. For this, we study TabCBM in our concept-annotated synthetic datasets, as they have ground truth binary concept annotations **c** and known ground truth concept feature masks $\{\pi^{(i)}\}_{i=1}^{k}$. We evaluate the alignment between TabCBM's learnt concept masks $\{\hat{\pi}^{(i)}\}_{i=1}^{k'}$ and those in $\{\pi^{(i)}\}_{i=1}^{k}$ by first computing an alignment $\alpha : \{1, 2, \cdots, k'\} \rightarrow \{1, 2, \cdots, k\}$ between discovered concepts and ground truth concepts. For the sake of efficiency, we compute this alignment by matching each learnt concept $\hat{c}_i$ with the ground truth concept $c_j$ whose training labels have the highest absolute Pearson linear correlation $|p_{(\hat{c}_i, c_j)}|$ with the training concept scores for $\hat{c}_i$. We then capture how well $\hat{\pi}^{(i)}$ is aligned with the feature mask of concept $c_{(\alpha(i))}$ using the *Mask AUC* (M-AUC) defined as M-AUC $:= \frac{1}{k'} \sum_{i=1}^{k'} \text{ROC-AUC}\big(\pi^{(\alpha(i))}, \hat{\pi}^{(i)}\big)$. Here, ROC-AUC$\big(\pi^{(\alpha(i))}, \hat{\pi}^{(i)}\big)$ is the area under the ROC curve when predicting ground truth $\pi^{(\alpha(i))}$ with the feature importance mask $\hat{\pi}^{(i)}$. This score is 1 if $\hat{\pi}^{(i)} = \pi^{(\alpha(i))}$ for all $i$ and 0 if $\hat{\pi}^{(i)}$ is the complement of $\pi^{(\alpha(i))}$ for all $i$.

Our results, summarised in Table 3, show that there is significant alignment between TabCBM's concept feature importance masks and ground truth concept feature masks. Nevertheless, we observe that in *Synth-scRNA*, TabCBM's concept masks are not as highly aligned as in other tasks. We believe that this is due to the high amount of noise found in this dataset, leading to TabCBM focusing on capturing only the most salient differentiating genes of each GEP.

Table 3: M-AUC and G-AUC scores and standard errors across synthetic datasets. For M-AUC we include only TabCBM as other concept-interpretable methods do not provide features for learnt concepts. All values are shown as percentages rounded up to two decimals (if an error is less than $10^{-3}\%$, then it is not shown).

| Score | Method | Synth-Linear | Synth-Nonlin | Synth-Nonlin-Large | Synth-scRNA |
|---|---|---|---|---|---|
| M-AUC | **TabCBM** | 100 | 100 | $85.71 \pm 5.60$ | $65.69 \pm 1.66$ |
| G-AUC | **TabCBM** | 100 | 100 | 100 | $69.01 \pm 2.61$ |
| | TabNet | $85.33 \pm 8.89$ | $75.33 \pm 4.42$ | $50.37 \pm 0.52$ | $50.39 \pm 0.15$ |
| | XGBoost | 100 | 100 | 100 | $56.10 \pm 0.5$ |
| | LightGBM | 100 | 100 | 100 | $56.11 \pm 0.52$ |

**Global feature selection**  Understanding which features are deemed redundant/unnecessary for a model is an important problem (Chandrashekar & Sahin, 2014; Lee et al., 2021; Imrie et al., 2022). In this section, we explore whether TabCBM's concepts can provide a sense of global feature importance for the downstream task via their mean feature important mask $\sum_i \hat{\pi}^{(i)}/k'$. We investigate this by designing a metric that, for any method that constructs a vector $\hat{\mu} \in \mathbb{R}^n$ indicating the relative importance of each input feature w.r.t. a downstream task, captures how well this vector predicts ground-truth important features for the task. With this aim, assume we have a binary ground truth vector $\mu \in \{0,1\}^n$ indicating which features are globally important (i.e., selected) w.r.t. a downstream task. We can compute how well $\hat{\mu}$ identifies these features using the *Global feature selection AUC* (G-AUC) defined as:

$$\text{G-AUC} := \max_{\delta \in S} \text{ROC-AUC}\Big(\mu, \text{threshold}\big(\hat{\mu}, \delta \max_j \hat{\mu}_j\big)\Big) \tag{8}$$

where $\text{threshold}(\mathbf{x}, \beta)$ is a vector function whose $i$-th output is given by $\mathbb{1}_{x_i \geq \beta}$. Similarly, we let $S \subset [0,1]$ be a finite set of reals indicating the thresholds used for binarising $\hat{\mu}$ (as a function of its maximum element) and we let $\text{ROC-AUC}(\mathbf{y}, \hat{\mathbf{y}})$ be the mean area under the ROC curve when predicting $\mathbf{y}_i$ from $\hat{\mathbf{y}}_i$ for all $i$. Intuitively, if there is a fraction $\delta \in S$ of $\hat{\mu}$'s maximum value that produces $\mu$ when thresholding $\hat{\mu}$, then the G-AUC is 100%. In contrast, a G-AUC of 0% indicates that $\hat{\mu}$ attains its maximum value in all features which are not selected in $\mu$. For efficiency purposes, in practice we use $S = \{0, 0.025, 0.05, \cdots, 1\}$.

For every synthetic dataset, we evaluate the G-AUC by computing $\mu_i := \bigvee_{j=1}^k \pi_i^{(j)}$ as an aggregated logical OR over all ground-truth masks. For fairness, we only compute the G-AUC for methods that provide feature importance by construction, therefore not requiring post-hoc methods (e.g., LIME (Ribeiro et al., 2016)) to obtain this mask. Hence, we focus on contrasting feature importance masks by XGBoost, LightGBM, TabNet, and TabCBM. For XGBoost and LightGBM we calculate $\hat{\mu}$ as the total gains of splits using each feature across all decision trees. Similarly, for TabNet, as in (Arık & Pfister, 2021), we compute $\hat{\mu}$ by aggregating all attention masks. Our results, shown in Table 3, indicate that TabCBM perfectly identifies all important features for our simpler synthetic tasks (as GBMs do too) while achieving significantly better G-AUC scores than other methods in *Synth-scRNA*, our more complex synthetic task. These results suggest that TabCBM can globally identify feature selection masks, allowing for global interpretability beyond that in current concept-learning methods. For further results showing how the value selected for $k'$ affects both the M-AUC and G-AUC metrics (as well as the task accuracy), see Appendix I.1.

## 5.4 Concept Test-time Interventions (Q4)

A significant advantage of the existence of a concept bottleneck in a CBM is to allow for concept-level interventions at test time. Through interactions with a CBM in a human-in-the-loop setting, an expert can improve the model's end performance by correcting mispredicted concepts and allowing the CBM to update its previously predicted downstream label to take the correction into account. In this section, we explore whether TabCBM allows for such a mechanism *even when no concept supervision* has been provided during training. For this, we begin with the reasonable assumption that a domain expert can identify strong linear correlations between ground-truth concepts (which are known by the expert) and learnt concept scores. Under this assumption, we propose an intervention procedure in TabCBM's bottleneck where, upon identifying that a learnt concept score is aligned with a known ground-truth concept, an expert can modify the predicted concept's score at test time to correct what they believe may be a misprediction, in turn possibly

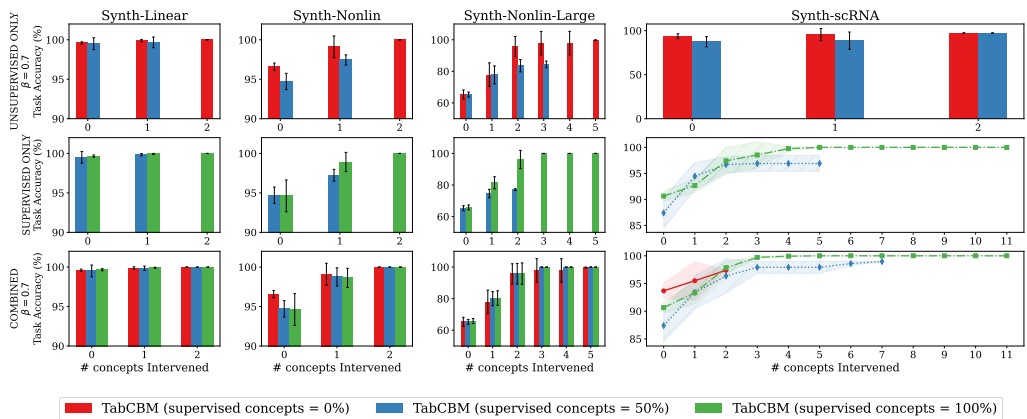

Figure 4: Task accuracy of TabCBMs when randomly intervening on a varying number of concepts (x-axis) across multiple datasets (columns). We show the effect of interventions on task accuracies when intervening only on learnt concepts that (i) were not provided with training supervision (top row), (ii) were provided with training concept supervision (middle row). The bottom row shows the results of intervening on discovered concepts *regardless of whether they received training supervision or not.*

affecting the TabCBM's downstream task prediction. Our results, discussed below, suggest that TabCBM can boost its performance through test-time interventions even when it receives no concept supervision.

To evaluate interventions in TabCBMs, we proceed as follows: we measure the Pearson linear correlation coefficient $p_{(i,j)}$ between every learnt concept score $\hat{c}_i$ and every known ground-truth concept $c_j$ in the domain dataset. If for a ground-truth concept $c_j$ we have that $\hat{c}_i$ has an absolute correlation coefficient greater than or equal to a pre-defined "alignment" threshold $\beta \in [0, 1]$ (e.g., we use $\beta = 0.7$), then we consider $\hat{c}_j$ to be aligned with $c_i$. In practice, an expert could identify such an alignment by inspecting concept scores in a subset of the training data and focusing on instances in which a learnt concept score reaches extreme values.

Let $\mathcal{A}_i := \{j \mid \beta \le |p_{(i,j)}|\}$ be the set of ground-truth concepts that are strongly aligned with learnt concept $\hat{c}_i$. Notice that this set may be empty if a concept score was not strongly aligned with any known concepts. When $\mathcal{A}_i$ is non-empty, let $\eta_i$ be the index $j \in \mathcal{A}_i$ of the ground-truth concept with the highest absolute correlation with learnt concept $\hat{c}_i$ (i.e., $\eta_i := \mathrm{argmax}_{j \in \mathcal{A}_i}(|p_{i,j}|)$). Given a set $\mathcal{I} \subseteq \{1, 2, \cdots, k'\}$ of learnt concepts we wish to intervene on, we perform test-time interventions by updating the bottleneck $\hat{\mathbf{c}}$ as follows:

$$\hat{c}_i := \begin{cases} \hat{c}_i & (\mathcal{A}_i = \emptyset) \text{ or } (i \notin \mathcal{I}) \\ \hat{c}_i^{[95\%]} c_{\eta_i} & (\mathcal{A}_i \ne \emptyset) \text{ and } (i \in \mathcal{I}) \text{ and } (p_{(i,\eta_i)} \ge 0) \\ \hat{c}_i^{[5\%]} (1 - c_{\eta_i}) & (\mathcal{A}_i \ne \emptyset) \text{ and } (i \in \mathcal{I}) \text{ and } (p_{(i,\eta_i)} < 0) \end{cases} \quad (9)$$

where $\hat{c}_i^{[95\%]}$ and $\hat{c}_i^{[5\%]}$ are the 95-th and 5-th percentiles of the empirical distribution of $\hat{c}_i$ in the training set, respectively. These values are used to represent a concept's activation or inactivation by making sure that we perform an intervention while maintaining the scores within their empirical distribution. Once this update has been performed, TabCBM updates its prediction by computing $f(\hat{\mathbf{c}})$ using the new concept score bottleneck. Notice that this update takes into account the fact that a concept score $\hat{c}_i$ may be positively or negatively correlated with a ground truth concept $c_j$ (i.e., TabCBM may learn the complement labelling of ground-truth concept $c_j$ by using $\hat{c}_i = 0$ to indicate $c_j$'s activation rather than $\hat{c}_i = 1$).

We evaluate this procedure on our synthetic datasets by intervening on a randomly selected set of concepts of varying size. These results are summarised in Figure 4. For the sake of completeness, we evaluate TabCBM's intervention effectiveness on concept-unsupervised and fully concept-supervised setups (partially concept-supervised setups look similar to fully-supervised and are included in Appendix H.1). Our results suggest that TabCBM is highly receptive to concept interventions both when it receives concept supervision as well *as when it lacks any form of concept supervision* (red solid bars/lines vs dashed coloured bars/lines). We can see this by observing the top row of Figure 4 where we show how TabCBM's performance improves significantly

when intervening only on unsupervised concepts. Only in the more complex dataset, namely our scRNA task, do we observe that interventions in unsupervised TabCBMs, although significantly beneficial to the model's end performance, are limited in number (i.e., on average we can intervene only on 1-2 unsupervised concepts in Synth-scRNA). This is because only a handful of discovered concepts were significantly linearly correlated with known ground truth concepts, and suggests that future work can explore how to exploit weaker non-zero correlations, as those seen in Figure 3, to better utilise partially aligned concepts in real-world tasks. Nevertheless, and perhaps more importantly, introducing unsupervised concepts in TabCBM can significantly improve its performance even when some of its learnt concepts are supervised. This can be seen by noticing that across all models, the performance of concept-supervised TabCBMs improves when we enable interventions in unsupervised concepts (see the improvement in the bottom row of Figure 4 compared to the middle row). These results strongly indicate that TabCBMs can benefit from being deployed in a human-in-the-loop setup, especially when a handful of high-level concepts are known (even if this set is not complete w.r.t. the downstream task). For further details on interventions, including a discussion on the effect of $\beta$ and an evaluation of TabCBM's interventions against those in CBMs and CEMs showing improvements against those baselines, see Appendix H.

## 6 Discussion

**Relation with Group Feature Selection** TabCBM's architectural principles are highly related to those in *group feature selection*, where methods learn multiple features subsets from which an ensemble can be built (Yuan & Lin, 2006; Zhou & Zhu, 2010; Imrie et al., 2022). Nevertheless, there are crucial differences between TabCBM and existing group feature selection methods. First, although as in group feature selection TabCBM uses subsets of highly correlated features to construct a highly predictive model for a downstream task, our method is capable of discovering such feature subsets without a priori knowledge required about the nature of these subgroups, as it is usually needed in traditional group feature selection methods. Second, TabCBM does not require the masks used for each discovered concept to be distinct from the masks of other discovered concepts, as multiple high-level concepts can be a function of overlapping subsets of concepts. This is in contrast to recent group feature selection methods such as Composite Feature Selection (Imrie et al., 2022). Third, our method allows for concept supervision to be provided at training time to encourage alignment between certain learnt concepts and known ground-truth concepts, something we have not found in previous literature on feature selection. Finally, TabCBM allows for test time interventions through modifications in its predicted concept scores, enabling better performance in a human-in-the-loop setting.

**Limitations and Future Work** Although we have shown that TabCBM achieves high accuracy in a variety of tasks while learning useful concept-based explanations, this comes with a set of limitations. First, as in topic models (Mcauliffe & Blei, 2007) and concept discovery methods (Yeh et al., 2020), TabCBM requires $k'$, the number of discovered concepts, to be a priori defined by a user. While selecting a reasonable value for $k'$ can be aided by domain knowledge or through a linear search over multiple values of $k'$, one is still required to fine-tune $k'$ to deploy TabCBM successfully. Luckily, as seen in our $k'$-ablation discussed in Appendix I.1, we empirically observe that TabCBM is relatively robust to changes in $k'$ as long as TabCBM is not under-parameterised. Nevertheless, future work can explore ways of automatically learning an optimal $k'$. Second, TabCBM can suffer from parameter and computational growth if either the number of concepts or the size of each concept embedding is large. Future work can explore alleviating this practical limitation through the use of parameter sharing between concept embedding generators and/or better transfer learning protocols which enable learning these generators independently (i.e., separately from the rest of the model). Third, given the lack of previous work on concept-based learning for tabular data, our interpretability evaluation was constrained to mostly synthetic datasets. This is because we found it difficult to find public tabular datasets with both concept and label annotations. Nevertheless, we hope that this work will encourage the application of TabCBM to tabular tasks where, when used with domain-specific experts, one may be able to discover novel properties which one may later use as annotations for training future concept-based models. Finally, our method requires several hyperparameters to be selected before training as it has several regularisers and architectural choices that need to be made a priori. This can lead to intensive hyperparameter searches and the requirement of access to certain hardware capabilities. To better understand TabCBM's sensitivity to its hyperparameters, we perform an ablation search over all our hyperparameters in Appendix I.2 and see

that our method can perform well within a large range of hyperparameters. To further help future research, we include a set of recommendations for each hyperparameter in that appendix.

## 7 Conclusion

In this paper, we address the lack of concept-based interpretable methods that apply to tabular tasks. We approach this by first arguing that non-linear functions of highly correlated subsets of input features can be thought of as a sensible definition of what a concept may entail in a tabular domain. Spanning from this definition, and combining multiple ideas from the fields of feature selection and concept learning, we propose TabCBM as the first concept-based interpretable neural architecture designed for tabular tasks. Our extensive evaluation of TabCBM, both in synthetic and real-world datasets, suggests that our model achieves competitive state-of-the-art predictive performance in a variety of tabular tasks whilst discovering concepts that align with known ground-truth concepts. We show that the concept representations learnt by TabCBM are disentangled, highly informative, and complete w.r.t. the downstream task. Furthermore, we show, both quantitatively and qualitatively, that TabCBM is capable of discovering meaningful concepts both when it is provided with partial concept supervision and when it is provided with no concept supervision at all. This enables our model to be deployed in a human-in-the-loop setup in which experts can easily identify and correct semantically meaningful concepts, leading to significant improvements in downstream task performance. This work, therefore, serves as an important stepping stone for designing high-performing interpretable tabular architectures, enabling the use of deep learning in high-stakes tabular tasks such as medical diagnosis where understanding a model's prediction is of utmost importance.

### Acknowledgements

The authors would like to thank Adrian Weller, Carl Henrik Ek, and Andrei Margeloiu for insightful discussions regarding this work and earlier versions of our manuscript. MEZ acknowledges support from the Gates Cambridge Trust via a Gates Cambridge Scholarship. MJ is supported by the EPSRC grant EP/T019603/1.

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

## A    Relation between Tabular Concepts and Subscale Scoring Models

We notice that our definition of a tabular concept is similar to the core idea behind interpretable subscale or additive scoring models (Chen et al., 2022; 2018a; Ustun & Rudin, 2016; 2017). The main differences between the two are that (1) we interpret each subscale as a high-level concept in that domain (whereas subscale models do not necessarily assign semantics to subcales), (2) we learn these subsets, and their corresponding functions, via supervision from a downstream task rather than using hand-crafted scoring scales, and (3) the function mapping each concept's features to its corresponding score can be highly non-linear (subscale models tend to assume linear functions between feature subsets and their scores).

## B    Multivariate Bernoulli Sampler Details

As indicated in Section 4, during training, we use TabCBM's learnt concept feature importance vectors $\{\hat{\pi}^{(i)}\}_{i=1}^{k'}$ to generate per-sample binary masks $\{\theta^{(i)}\}_{i=1}^{k'}$ where each feature's probability of being selected by $\theta^{(i)}$ is proportional to their corresponding feature importance in $\hat{\pi}^{(i)}$. Furthermore, we avoid assuming full independence between features when generating $\theta^{(i)}$ from $\hat{\pi}^{(i)}$ by considering $\hat{\pi}^{(i)}$ only as the marginal probability distributions of selecting each feature and introducing a non-trivial covariance matrix in the sampling process that considers inter-feature correlations. As argued by Lee et al. (2021), this has two advantages: (i) during the pre-training stage of the feature importance masks (see Appendix C), this process

encourages our latent code generator to avoid relying on information leaked from highly correlated unmasked features to solve the self-supervised pretext tasks, therefore improving the quality of pre-trained learnt representations, and (ii) using a non-trivial covariate matrix to sample $\theta^{(i)}$ encourages TabCBM's concepts to select features that are highly correlated together, aligning with our definition of tabular concepts.

In practice, this is achieved by sampling $\theta^{(i)}$ from a multivariate Bernoulli distribution we construct using a Gaussian copula (Nelsen, 2007). A Gaussian copula is a multivariate cumulative distribution function (CDF) for random variables $\{X^{(l)}\}_{l=1}^{n}$ in $[0,1]^n$ whose marginals are uniformly distributed (i.e., $X^{(l)} \sim \mathrm{Unif}(0,1)$). Given a matrix $\mathcal{C} \in [-1,1]^{n \times n}$ capturing the correlations of random variables $\{X^{(l)}\}_{l=1}^{n}$, we define the Gaussian Copula as:

$$\mathrm{Copula}_{\mathcal{C}}(X^{(1)}, \cdots, X^{(n)}) := \Phi_{\mathcal{C}}(\Phi^{-1}(X^{(1)}), \cdots, \Phi^{-1}(X^{(n)}))$$

where we use $\Phi^{-1}$ to indicate the inverse CDF of a standard univariate normal distribution and $\Phi_{\mathcal{C}}$ to indicate the joint CDF of a multivariate zero-mean normal distribution with correlation matrix $\mathcal{C}$. This construction is extremely powerful as it allows us to sample a binary mask vector $\theta^{(i)}$ from the multivariate Bernoulli distribution $\mathrm{MultiBern}(\hat{\pi}^{(i)}; \mathcal{C})$ using the copula's correlated random variables $\{X^{(l)}\}_{l=1}^{n}$ by letting

$$\theta_l^{(i)} := \begin{cases} 1, & \text{if } X^{(l)} \leq \hat{\pi}_l^{(i)} \\ 0, & \text{otherwise} \end{cases}$$

In TabCBM we construct this copula by building $\mathcal{C} \in [-1,1]^{n \times n}$ via the empirical feature Pearson correlation coefficients computed in the task's training set.

Finally, as in (Lee et al., 2021), we make this sampling process differentiable by working with a relaxation of the multivariate Bernoulli distribution proposed by Wang & Yin (2020) and using the reparameterisation trick on the copula's uniform random variables $\{X^{(l)}\}_{l=1}^{n}$. Specifically, given $\mathrm{Copula}_{\mathcal{C}}$'s uniform random variables $\{X^{(l)}\}_{l=1}^{n}$ and the marginal selection probabilities as dictated by a concept's feature importance mask $\hat{\pi}^{(i)}$, we can generate relaxed masking vector $\theta^{(i)}$ by setting it to

$$\theta_l^{(i)} := \sigma\Big(\frac{1}{\tau}\big(\log \hat{\pi}_l^{(i)} - \log(1 - \hat{\pi}_l^{(i)}) + \log X^{(l)} - \log(1 - X^{(l)})\big)\Big)$$

where $\sigma$ is the sigmoid function and $\tau \in (0, \infty)$ is a temperature hyperparameter which we fix to 1 in our experiments. This allows the output relaxed mask $\theta^{(i)}$ to be differentiable with respect to $\hat{\pi}^{(i)}$, therefore enabling us to learn each concept's feature importance vectors via gradient descent. In practice, we implement this relaxation using a deterministic transformation of a standard Gaussian noise vector whose entries encode the correlation structure in the features after being projected into the subspace $L$ of the lower-triangular Cholesky decomposition $LL^T$ of $\mathcal{C}$. For further details, see Section 4.3 of (Lee et al., 2021).

## C   TabCBM pre-training

In practice, we speed up TabCBM's learning via two pre-training steps: In the first step, we encourage the latent code generator $\phi(\mathbf{x})$ to learn meaningful representations by training it to minimise the expected task loss $\mathbb{E}_{(\mathbf{x},y) \sim \mathcal{D}}[\mathcal{L}_{\text{task}}(g_{\text{downstream}}(\phi(\mathbf{x})), y)]$, where $g_{\text{downstream}} : \mathbb{R}^m \to [0,1]^L$ is a learnable auxiliary linear layer used to predict a class $\hat{y}$ from $\phi(\mathbf{x})$. This encourages $\phi$ to learn representations that are meaningful for the downstream task.

In a second pre-training step, we use self-supervision to learn meaningful latent code generators using the pre-text task proposed by Yoon et al. (2020) and also used in (Lee et al., 2021). Specifically, we first randomly initialise each concept feature importance mask $\hat{\pi}^{(i)}$ by sampling it from $\mathrm{Unif}(0.4, 0.6)$. We chose to sample each initial mask from $\mathrm{Unif}(0.4, 0.6)$ rather than from $\mathrm{Unif}(0,1)$, as we found that conditioning their values to be closer to 0.5 at the start of training helped with the training dynamics. For each training sample $\mathbf{x}$ and each learnt concept $i$, we then sample a binary mask $\theta^{(i)} \in \{0,1\}^n$ from a multivariate Bernoulli distribution whose marginal probabilities are fixed to $\hat{\pi}$ and covariance matrix is captured via $\mathrm{Copula}_{\mathcal{C}}$. As opposed to how we proceed in our training stages for TabCBM, in this pre-training stage we do not apply any relaxations

when generating $\theta^{(i)}$, meaning that it will be binary rather than continuous as in our training stages and that we will not update any feature importance vectors $\hat{\pi}^{(i)}$. This is done for us to be able zero out elements of input $\mathbf{x}$ and produce a masked vector $\tilde{\mathbf{x}}$ from which we train $\phi(\tilde{\mathbf{x}})$ so that the latent code obtained from masked input $\tilde{\mathbf{x}}$ is informative enough for an auxiliary model $g_{\text{mask}} : \mathbb{R}^m \to [0, 1]^n$ to be able to predict mask $\hat{\theta}$. Simultaneously, we also condition our learnt latent code to be a meaningful representation by using it as the input of a separate simple auxiliary model $g_{\text{entries}} : \mathbb{R}^m \to \mathbb{R}^n$ from which we aim to closely reconstruct the original sample $\mathbf{x}$ from $\phi(\tilde{\mathbf{x}})$. Both of these objectives can be achieved by finding the parameters for $\phi$, $g_{\text{mask}}$, and $g_{\text{entries}}$ that minimize the weighted loss independently for each concept

$$\frac{1}{k'} \sum_{i=1}^{k'} \mathbb{E}_{\mathbf{x} \sim \mathcal{D}, \theta^{(i)} \sim \text{MultiBern}(\hat{\pi}^{(i)}; \mathcal{C})} \left[ \mathcal{L}_{\text{cross-entropy}}(g_{\text{mask}}(\phi(\mathbf{x} \odot \theta^{(i)})), \theta) + \lambda_{\text{rec}} ||g_{\text{entries}}(\phi(\mathbf{x} \odot \theta^{(i)})) - \mathbf{x}||_2^2 \right]$$

where $\lambda_{\text{rec}} \in \mathbb{R}^+$ is a hyperparameter controlling how much we value perfect reconstruction over mask identification (we fix it to $\lambda_{\text{rec}} = 10$ throughout evaluation). The purpose of this pre-training step is to precondition the latent code generator $\phi(\cdot)$ to capture meaningful semantics of the input features when some of its features are masked, as this will be done when learning different concepts for TabCBM.

For all of our experiments, we pre-train TabCBM by running the first pre-training stage for $T_{\text{pre-train}}$ epochs and then running the second pre-training stage for $T_{\text{self-supervised}}$ epochs (specific parameters are described in Section E.2). Furthermore, we use a linear layer for the helper model $g_{\text{downstream}}$ and a ReLU MLP with a single hidden layer size of 64 for the $g_{\text{mask}}$ and $g_{\text{entries}}$ helper models. Finally, for the sake of fairness, we pre-train the backbones of all other competing models using the first stage of this procedure for $T_{\text{pre-train}} + T_{\text{self-supervised}}$ epochs to allocate equivalent computational budgets across all methods.

## D  Datasets

In this section, we give a detailed description of the datasets we use for our evaluation of TabCBM. A summary of each dataset's core characteristics can be found in Table 4.

Table 4: Characteristics of all tasks used in this paper. Notice that we worked exclusively with classification tasks, so all downstream labels are categorical. The *class imbalance ratio* is defined as the ratio between the number of samples of the most represented class and the number of samples of the least represented class.

| Dataset | # of Samples | # of Features | # of Concepts | # of Classes | Class Imbalance Ratio |
|---|---|---|---|---|---|
| Synth-Linear | 15,000 | 100 | 2 | 4 | 1.05 |
| Synth-Nonlin | 15,000 | 100 | 2 | 4 | 1.05 |
| Synth-Nonlin-Large | 15,000 | 500 | 5 | 32 | 7.4 |
| Synth-scRNA | 7,500 | 5,000 | 11 | 20 | 2.6 |
| Higgs (without high-level) | 11,000,000 | 21 | n/a | 2 | 1.13 |
| Higgs (with high-level) | 11,000,000 | 28 | n/a | 2 | 1.13 |
| PBMC | 20,738 | 6,212 | n/a | 2 | 1.01 |
| FICO | 10,459 | 23 | n/a | 2 | 1.09 |

**Synthetic Tabular Datasets**  To evaluate our methods in a controlled environment, we propose three synthetic tabular tasks with increasing difficulty. The first two tasks, namely *Synth-Linear* and *Synth-Nonlin*, consist of $N = 15,000$, 100-dimensional samples $\{\mathbf{x} \in \mathbb{R}^{100}\}_{i=1}^N$ generated from a corresponding latent vector $\mathbf{h} \in \mathbb{R}^{100} \sim \mathcal{N}(\mathbf{0}, I)$, where $I$ is the identity matrix (each sample $\mathbf{x}$ has its own corresponding latent vector $\mathbf{h}$). Then, we generate each sample $\mathbf{x}$ from its corresponding latent vector $\mathbf{h}$ by applying a transformation $f : \mathbb{R} \to \mathbb{R}$ to $\mathbf{h}$. In the case of *Synth-Linear*, we let $f(\mathbf{h}) = \mathbf{h}$ be the identity function while in *Synth-Nonlin*, we let $f(\mathbf{h}) = \sin(\mathbf{h}) + \mathbf{h}$ be a non-linear function of $\mathbf{h}$. To assign a label $y$ to each sample, we first annotate each sample with a *binary concept vector* $\mathbf{c} \in \{0, 1\}^k$, with $k = 2$ for these two datasets, such that its $j$-th dimension $c_j := \mathbb{1}_{(\sum_{l=js+1}^{(j+1)s+1} h_l) \geq 0}$ indicates the sign of adding a subset of $s$ consecutive features in the latent vector $\mathbf{h}$ (we set the spacing $s$ to be 5 for these two datasets). This binary vector is then used to produce a label $y \in \{0, 1, \cdots, 2^k - 1\}$ by generating $y$ from the decimal representation of vector $\mathbf{c}$ i.e.,

$y = \left(\overline{c_1 c_2 \cdots c_k}\right)_{10}$. Notice then that the label assigned to each sample can be uniquely determined from its corresponding concept vector $\mathbf{c}$, which, in turn, is a function of only the first $sk$ dimensions of sample $\mathbf{x}$. Therefore, to properly learn the generative process for these datasets, a method must be able to capture the importance of the first $sk$ dimensions and discover, either implicitly or explicitly, the concept vector $\mathbf{c}$.

We further extend the generative process of *Synth-Nonlin* and generate a more complex dataset, which we refer to as *Synth-Nonlin-Large*, in which features have 500 dimensions rather than 100 and $k = 5$ binary concepts $\mathbf{c} \in \{0,1\}^5$ are constructed to determine label $y \in \{0, 1, \cdots, 2^5 - 1\}$ instead of only two concepts. As in *Synth-Nonlin*, each dimension $c_j$ of $\mathbf{c}$ is computed using an indicator variable of a subset of features in $\mathbf{h}$. In contrast with *Synth-Nonlin*, however, we allow different dimensions of $\mathbf{c}$ to depend on overlapping sets of features in $\mathbf{h}$. We do this by letting $c_j$ be defined as $c_j := \mathbb{1}_{\left(\sum_{l=\max(js-s/2-1,1)}^{(j+1)s+s/2+1} h_l\right) \geq 0}$ and setting $y = \left(\overline{c_1 \cdots c_5}\right)_{10}$ as we did for *Synth-Nonlin*. For this dataset, we use a spacing factor of $s = 20$ to construct our concepts, given that we also have more features than in the previous two datasets. This means that only the first 100 features of the dataset will be useful for learning the downstream task.

**Synth-scRNA**  As an exploration of applying TabCBM to meaningful bioinformatics tasks, we explore a synthetic RNAseq (Macaulay & Voet, 2014) task where each sample contains gene expression counts for a total of $5,000$ genes at a single-cell resolution. To generate such a dataset, we follow the approach suggested by Kotliar et al. (2019) and adapt *Splatter* (Zappia et al., 2017) to generate a sc-RNAseq dataset where we can control the number of identity and activity gene expression programs (GEPs) (Segal et al., 2003) in the cell population. In this context, *identity GEPs* are aligned in a one-to-one fashion with different cell types and represent groups of signature genes that are co-regulated for each cell type differentiation. In contrast, *activity GEPs* represent groups of co-regulated genes which are not necessarily related to a cell's type and instead represent biological functions related to environmental/epigenetic effects (e.g., genes related to different stages of a cell's cycle). For our experiments, we generate a synthetic RNAseq dataset with 7,500 samples in it, with ten different cell types (i.e., ten identity GEPs) and one *activity GEP* which *can be expressed across all cell types*. Given that being able to disentangle and learn identity and activity GEPs from sc-RNAseq data is an important task in genomics (Macaulay & Voet, 2014; Kotliar et al., 2019; Kharchenko et al., 2014; Satija et al., 2015), this synthetic task is representative of a potentially impactful application of our algorithm.

In practice, we generate this dataset using Splatter's Python implementation by Kotliar et al. (2019). In this framework, we call Splatter by setting the number of genes to $5,000$, the number of cells to $7,500$, and the number of genes differentiating in the task's activity GEP to be 250 (we extended Splatter's implementation by Kotliar et al. (2019) to allow for a specific number of genes used for each GEP). All other Splatter parameters (e.g., `lib.loc`, `lib.scale`, etc), except for the number of doublets which we set to 0 (i.e., we do not simulate doublets), are set to the same values as those used for the synthetic scRNA-seq dataset in (Kotliar et al., 2019). These values were estimated using Splatter on a subset of the organoid dataset proposed by Quadrato et al. (2017).

Before training, we preprocess the generated dataset by (1) removing all cells with less than 200 gene expression counts, (2) removing all cells with less than 200 genes with non-zero counts in total, (3) removing all genes expressed in less than 50 cells, and (4) normalising all counts for all cells (normalisation is done on a per-cell basis) and transforming these normalised counts to a log space using a $log(x + 1)$ transformation (as commonly done for scRNA-seq data (Kotliar et al., 2019)).

To produce a task in which GEPs are relevant for the class, we assign each cell a label representing its cell type (which is in one-to-one correspondence with an identity GEP) and whether or not the dataset's activity GEP is on. This yields an annotated dataset where each sample has one of $10 \times 2 = 20$ labels and each label can be uniquely determined by knowing the cell's identity and activity GEPs (see Figure 5 for a visualisation of this dataset). Therefore, in this task both identity and activity GEPs can be thought of as ground-truth high-level concepts which can fully describe the label in our task of interest. Furthermore, as we know the genes that are differentially expressed for each GEP, we can evaluate how accurately our method can discover the set of differentiating genes for each GEP using these masks as ground-truth concept masks.

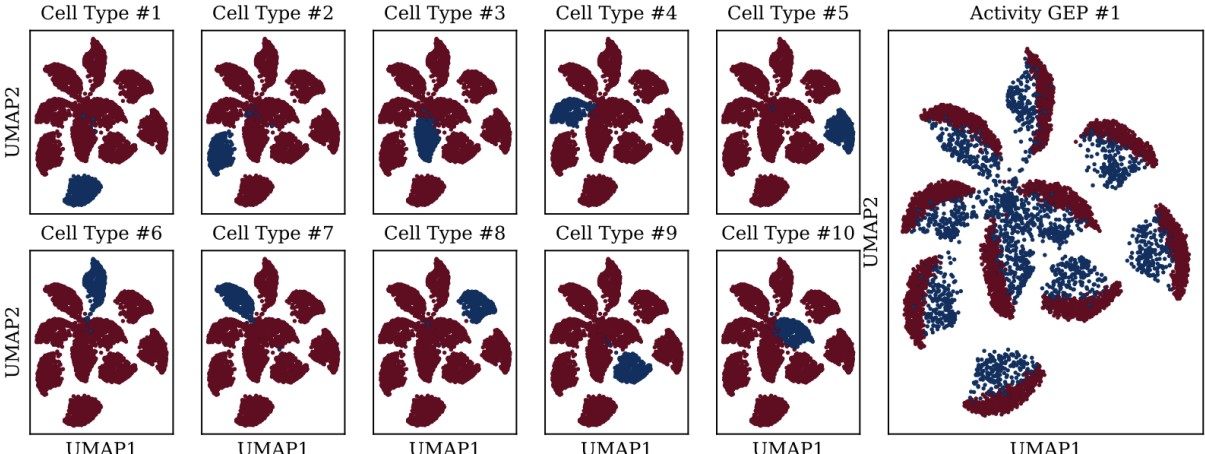

Figure 5: UMAP visualisations of the Synth-scRNA dataset with each identity GEP (i.e., cell type) and activity GEP highlighted.

**PBMC**    We further explore an application of our method in a real-world single-cell transcriptomics dataset. With this aim, and inspired by the work by Lee et al. (2021), we construct a binary transcriptomics task from two subpopulations of peripheral blood monocytes obtained from the PBCM (10x Genomics, 2016a;b) database. Specifically, we construct a dataset with approximately 20,000 samples where each sample contains transcriptomics measurements of nearly 6,000 genes of a T-cell (i.e., RNA expression patterns). We preprocess and normalised all counts as done for our Synth-scRNA gene counts and annotate each cell with a binary task label describing whether the cell corresponds to a regulatory or a naive T-cell.

**Higgs**    As a real-world example of a large tabular dataset in the physical sciences, we make use of the *Higgs* dataset (Baldi et al., 2014). In this dataset, we aim to classify a signal process as being one from which a Higgs boson is produced or as being a background signal. All 11 million samples in this dataset were produced via a Monte Carlo simulation and each sample signal is formed by 28 continuous features. Of these features, the first 21 features represent kinematics properties whereas the latter 7 features are high-level features hand-crafted by experts which allow easier discrimination between background and Higgs-producing signals. To explore whether hierarchical models such as TabCBM can learn the information encoded into the hand-crafted features without extra supervision, we evaluate all models both when using the 7 hand-crafted high-level features (*Higgs (with high-level)*) and when none of the hand-crafted high-level features are used (*Higgs (without high-level)*).

**FICO**    Given that a large proportion of tabular data is seen within financial tasks, and the fact that these tasks include both numerical and categorical features, we include the Fair Isaac Corporation (FICO) loan default risk challenge dataset (Fair Isaac Corporation, 2019) as part of our evaluation. We choose this specific dataset as it has been shown that simple handcrafted subscale models, whose relationship with high-level concepts was not made until this work, can be used to construct interpretable classifiers for this dataset (Chen et al., 2018a). Each of the approximately 10,000 samples in this dataset is composed of 23 features, a mixture of numerical and categorical features representing several properties of a client's past loan and financial history, and is annotated with a label indicating whether this client defaulted on their loan. The task is to predict the likelihood of a new client defaulting on a new loan.

## E    Experimental Details

In this section, we provide details on each method's configuration and hyperparameters for each task evaluated in this paper. All of our numerical results are produced by *averaging a given metric across 5 different random initialisations of each model* and showing the *standard error* found across those runs.

### E.1 Training and Optimisation Hyperparameters

For each method, and across all tasks, we split each task's dataset into 80% training data and 20% test data and generate a validation set by randomly sampling without substitution 20% of the training data. When sensible, this validation set is used to search over a subset of possible hyperparameters for each method by running a simple grid search over a set of predefined hyperparameters (described when appropriate below). Once the best hyperparameters are selected, we report only on the test results obtained for the model with the best validation error.

As is commonly done for training DNNs, we train all neural models via stochastic gradient descent with the same batch size $B$ for all methods across a given task. We choose a specific batch size to maximise GPU utilisation while remaining within our hardware's memory capabilities. We used an Adam optimiser (Kingma & Ba, 2014) with learning rate $10^{-3}$, momentum 0.99, and standard hyperparameters $\beta_1 = 0.9$ and $\beta_2 = 0.999$, across all methods and tasks. The exception for this is TabNet, where the learning rate starts at 0.02 and decays with a factor of 0.9 every ten steps as suggested by Arık & Pfister (2021). All methods are trained for $T$ epochs, stopping early (Prechelt, 1998) if no improvements have been observed in the validation loss for $P$ epochs. Hyperparameter values used for each dataset are reported in Table 5.

### E.2 Model Selection

For the sake of fairness, we provide each method with the same capacity and training times when possible. With this aim in mind, we fix the architecture used across methods to be the same for a given dataset. We select architectures that are simple to train, yet large and expressive enough to perform well in each task of interest; with the constraint that they should train in our GPU cluster within reasonable times. In this subsection, we list the architectures used for all methods across a given dataset and summarise their exact values used for all datasets in Table 5.

**TabCBM Model Selection**   Across all datasets, we construct TabCBM's latent representation generator using a simple ReLU MLP whose hidden layers have sizes $\phi_{\text{hidden-sizes}}$ and whose output layer has size $m$ (with $m$ being the size of the learnt concept embeddings). To help reduce training-time covariance shifts, we also introduce a batch normalisation (Ioffe & Szegedy, 2015) layer before each linear layer of this MLP, including the first layer. We then construct TabCBM's label predictor $f$ using a ReLU MLP with hidden layers whose sizes are determined by $f_{\text{hidden-sizes}}$. The number of output activations of $f$ will correspond to the number of output classes for the task of interest. Across *all* tasks, we fix the architectures of TabCBM's $\rho^{(i)}$ models to be ReLU MLPs with a single hidden layer with size 64 for all $i \in \{1, \cdots, k'\}$ (the only exception for this are the FICO dataset where, due to hardware constraints, we reduce this to 32 and the PBMC task where we use an MLP with units $\{32, 16\}$ to avoid parameter growth). Moreover, when the number of ground truth concepts is unknown, we train TabCBMs with $k' \in \mathcal{K}$ and select the model with the lowest validation loss. Otherwise, we use $k' = k$, where $k$ is the number of known ground truth concepts in the dataset. As for the loss term weights, we select the values for $\lambda_{\text{co}}$, $\lambda_{\text{div}}$, and $\lambda_{\text{spec}}$ via a grid search (by training for a few epochs only) as we vary each parameter, independently of each other, over $\{0.1, 1, 5, 10\}$. Finally, we set $t$, the hyperparameter of our coherence loss, to $t = (\text{batch size} \cdot \text{average class ratio})/2$ as suggested by Yeh et al. (2020). We report the best-performing parameters for each task in Table 5.

**MLP Model Selection**   For simplicity, we construct our *MLP baseline* by constructing a ReLU MLP whose hidden layers have size $\phi_{\text{hidden-sizes}} \cup \{m\} \cup f_{\text{hidden-sizes}}$ and where a batch normalisation layer is introduced before the first $\phi_{\text{hidden-sizes}}$ hidden layers, as in TabCBM. To ensure a fair comparison against TabCBM, which has more parameters than this MLP due to its concept-generating models, we also consider a larger-capacity MLP which has twice as many activations in each layer as those defined by $\phi_{\text{hidden-sizes}} \cup \{m\} \cup f_{\text{hidden-sizes}}$. However, in practice, we observe that this did not bring noticeable performance benefits (in fact, it sometimes even decreases performance, possibly due to overfitting), and therefore we opted for the smaller MLPs across all datasets.

**CCD Model Selection**   For CCD, we construct concept-based explanations for the MLP model defined above using the latent representation at layer $|\phi_{\text{hidden-sizes}} \cup \{m\}|$ as the representation we will aim to recon-

struct using CCD's learnt concept scores. Its $K$ hyper-parameter (representing how many top-K neighbours we use to compute its coherency regulariser) is set to $K = (\text{batch size} \cdot \text{average class ratio})/2$, as suggested by the authors, and the thresholding hyperparameter $\beta$, indicating the activation of a given concept, is fixed to 0. Similarly, we fix CCD's hyperparameters $\lambda_1$ and $\lambda_2$ to $\lambda_1 = \lambda_2 = 0.1$ after experimenting with values in $[0.01, 1]$ and finding these values to work well across the validation sets. Finally, for the reconstructing model $g$ of CCD's topic model (this is the model that will take the concept scores as input and attempt to reconstruct the latent representation of the DNN), we use a ReLU MLP with a single hidden layer with 500 activations in it and whose output layer has $m$ activations.

**SENN Model Selection**   For our evaluation of SENN, we begin by constructing an autoencoder $\phi_{\text{SENN}}$ whose encoder uses the same architecture as TabCBM's $\phi$ function with the exception that its output layer generates $k'$ activations (selected in the same way as in TabCBM). The decoder of this autoencoder is selected to be a ReLU MLP whose hidden layers have size $f_{\text{hidden-sizes}}$ and whose output is the same size as the number of input features in the task. This autoencoder is trained by minimising its reconstruction loss for $T_{\text{pre-train}} + T_{\text{self-supervised}}$ epochs. We then train a SENN using $\phi_{\text{SENN}}$ as its concept encoder and a ReLU MLP with hidden layer sizes $\theta_{\text{hidden-units}}$ and $L \times k'$ outputs (one vector of size $k'$ for each output class $\{1, \cdots, L\}$) as the relevance parameteriser. Throughout training, we follow the suggestions by Alvarez-Melis & Jaakkola (2018) and fix SENNs regularisation strength to $\lambda = 0.1$ and its sparsity regularisation strength to $\zeta = 2 \times 10^{-5}$.

**TabNet Model Selection**   For TabNet we explore different architectures with different capacities. We do this by exploring setting hyperparameters $N_d$, $N_a$, $N_{\text{steps}}$ by iterating their values from a set of options in sets $\mathcal{N}_d = \mathcal{N}_a = \{8, 16, 32, 64, 96\}$ and $\mathcal{N}_{\text{steps}} = \{3, 5, 8\}$, respectively. Nevertheless, as with MLPs, we observe that over-parameterising TabNet led to worse-performing models, therefore in Table 5, we only report the values of these hyperparameters that resulted in the best validation error. The virtual batch size of all TabNet models is set to half of the batch size for our dataset (i.e., $B/2$). Furthermore, we pre-train TabNet, using the self-supervised task described by Arık & Pfister (2021), for $T_{\text{pre-train}} + T_{\text{self-supervised}}$ epochs.

**TabTransformer Model Selection**   When constructing a TabTransformer model, we use the default parameters defined by Huang et al. (2020a) while setting the number of hidden layers to $\mathcal{T}_{\text{hidden-layers}}$.

**XGBoost and LightGBM Model Selection**   For both of our gradient boosting baselines, namely XGBoost and LightGBM, we train them for $T$ epochs, with the same early stopping patience as our neural baselines, while we vary the max depth of each tree between $\{5, 10\}$. All other hyperparameters are left to the default values suggested by their respective official implementations. In Table 5, we show the depth $d$ of the model achieving the best validation accuracy.

Table 5: Parameters used to generate the different baseline architectures for different datasets.

| Param | Synth-Linear | Synth-Nonlin | Synth-Nonlin-Large | Synth-scRNA | Higgs (with high) | Higgs (no high) | PBMC | FICO |
|---|---|---|---|---|---|---|---|---|
| $B$ | 1024 | 1024 | 1024 | 1024 | 2048 | 2048 | 1024 | 2024 |
| $T$ | 1,500 | 1,500 | 1,500 | 3,000 | 1,500 | 1,500 | 1,500 | 1,500 |
| $P$ | 250 | 250 | 250 | 750 | 50 | 50 | 250 | 300 |
| $T_{\text{pre-train}}$ | 50 | 50 | 50 | 50 | 25 | 25 | 25 | 50 |
| $T_{\text{self-supervised}}$ | 50 | 50 | 50 | 50 | 25 | 25 | 25 | 50 |
| $\phi_{\text{hidden-sizes}}$ | $\{16, 16\}$ | $\{16, 16\}$ | $\{16, 16\}$ | $\{128, 64, 64\}$ | $\{1024, 512, 256, 128\}$ | $\{1024, 512, 256, 128\}$ | $\{128, 64, 32\}$ | $\{256, 128, 64, 64\}$ |
| $m$ | 16 | 16 | 16 | 16 | 64 | 64 | 64 | 64 |
| $f_{\text{hidden-sizes}}$ | $\{16\}$ | $\{16\}$ | $\{16\}$ | $\{64, 32\}$ | $\{32, 16\}$ | $\{32, 16\}$ | $\{16, 16\}$ | $\{16\}$ |
| $\mathcal{K}$ | $\{2\}$ | $\{2\}$ | $\{5\}$ | $\{11\}$ | $\{4, 6, 8\}$ | $\{4, 6, 8\}$ | $\{3, 4, 5, 6\}$ | $\{4, 6, 8\}$ |
| $k'$ | 2 | 2 | 5 | 11 | 8 | 8 | 6 | 4 |
| $\lambda_{\text{co}}$ | 0.1 | 0.1 | 0.1 | 1 | 0.1 | 0.1 | 0.1 | 0.1 |
| $\lambda_{\text{div}}$ | 5 | 5 | 5 | 10 | 0.1 | 0.1 | 0.1 | 10 |
| $\lambda_{\text{spec}}$ | 5 | 5 | 5 | 10 | 0.1 | 0.1 | 0.1 | 10 |
| $\theta_{\text{hidden-units}}$ | $\{16, 16\}$ | $\{16, 16\}$ | $\{16, 16\}$ | $\{64, 64\}$ | $\{300, 300\}$ | $\{300, 300\}$ | $\{300, 300\}$ | $\{300, 300\}$ |
| $N_d$ | 8 | 8 | 64 | 64 | 32 | 96 | 16 | 32 |
| $N_a$ | 8 | 8 | 64 | 64 | 32 | 32 | 16 | 32 |
| $N_{\text{steps}}$ | 3 | 3 | 5 | 5 | 5 | 8 | 3 | 5 |
| $\mathcal{T}_{\text{hidden-layers}}$ | $\{16, 16, 16\}$ | $\{16, 16, 16\}$ | $\{16, 16, 16\}$ | $\{64, 32\}$ | $\{1024, 512 \cdots, 16\}$ | $\{1024, 512 \cdots, 16\}$ | $\{16, 16, 16, 16, 16\}$ | $\{32, 32, 32, 16, 16\}$ |
| $d_{\text{XGBoost}}$ | 5 | 5 | 5 | 5 | 10 | 10 | 5 | 5 |
| $d_{\text{LightGBM}}$ | 5 | 5 | 10 | 5 | 10 | 10 | 5 | 5 |

## F   Predictive Performance Experiment Details

When evaluating the concept-supervised baselines (i.e., TabCBM, CBM, Hybrid-CBM, and CEM), we vary the number of concepts provided during training as well as the fraction of training samples $r := \frac{|\mathcal{D}_{\text{concept-sup}}|}{|\mathcal{D}|}$ that have concept annotations in them. When subsampling only $k_{\text{selected}}$ concepts out of a total of $k$ available concepts in a dataset (e.g., $k = 2$ for *Synth-Linear* and *Synth-Nonlin* while it is 5 and 11 for *Synth-Nonlin-Large* and *Synth-scRNA*, respectively), we select a subset of concepts $C' \subset \{1, 2, \cdots, k\}$ of size $|C'| = k_{\text{selected}}$, chosen uniformly at random from all subsets of size $k_{\text{selected}}$, before training begins. We then provide all concept-supervised methods with concepts annotations $\mathbf{c} \in \{0, 1\}^{k_{\text{selected}}}$ for a fraction $r \in [0, 1]$ of the training samples, where the entries in $\mathbf{c}$ correspond to the concepts selected in $C'$. The hyperparameter $r$ decides the probability that a sample $(\mathbf{x}, y)$ in the training set has annotations $\mathbf{c}$. If a sample is not provided with concept annotations, then we set its contribution to the concept-specific loss to zero.

In terms of models used, for CBMs, Hybrid-CBMs, and CEMs we use the same architecture as TabCBM's $\phi$ model for their concept encoders (or latent code generator in case of CEM). Similarly, we use the same architecture as TabCBM's $f$ model as the label predictor for all of these models, with the exception that its input layer's size may be altered for CBMs if they are provided with fewer concepts than $k$. Similarly, $\phi$'s output layer's size is extended for CEMs so that it matches CEM's bottleneck size given by $k \times m_{\text{CEM}}$ (with $m_{\text{CEM}}$ being the embedding size of CEM's embeddings). Throughout all of these experiments, we keep the number of concepts discovered by TabCBM to $k' = k$ regardless of how many concepts $0 \le k_{\text{selected}} \le k$ were given at training time. Notice, however, that this implies that CBM's bottleneck size will always be $k_{\text{selected}}$, which can be highly constrained when $k_{\text{selected}}$ is small (a key limitation of CBMs). Nevertheless, for the sake of providing a fair evaluation against a CBM-like model and TabCBM, we compare against Hybrid-CBMs whose bottleneck sizes will always be fixed to $k'$, where $k' - k_{\text{selected}}$ of its activations will be unsupervised. This enables Hybrid-CBMs to have as much capacity in their bottlenecks as TabCBMs. Finally, we use a sigmoidal activation for both the bottleneck of CBMs and Hybrid-CBMs, as done in (Koh et al., 2020b) and (Espinosa Zarlenga et al., 2022), and we set CEM's embedding size to 8 and its RandInt parameter to $p_{\text{int}} = 0.25$ as suggested by Espinosa Zarlenga et al. (2022). For all of these models, including TabCBM, we fix the weight of the concept loss to be $\lambda_{\text{concept-sup}} = 5$ after trying values in $\{0.1, 1, 5, 10\}$ and finding that the best validation results were obtained with $\lambda_{\text{concept-sup}}$ in the range $(1, 10)$.

## G   Extra Qualitative Results in Synth-scRNA

As done for TabCBM in Section 5.2, we plot the same randomly selected GPEs in *Synth-scRNA* as in Figure 3 and show the scores of concepts learnt by CCD and SENN that are most linearly correlated with those GEPs. Our results, shown in Figures 6a and 6b, highlight that the alignment we observe in Figure 3 between TabCBM's discovered concept scores and ground-truth GEPs is not found for concepts learnt by SENN and CCD. We reiterate that TabCBM was able to attain the alignment shown in Figure 3 *without any concept supervision*. This indicates that, in contrast with SENN and CCD whose concepts do not seem to align to what one would expect in the *Synth-scRNA* task, TabCBM can discover concepts that align with human-interpretable high-level concepts for the downstream task of interest.

## H   Concept Intervention Experiments

In this section, we complement our results in Section 5.4 by further exploring interventions in TabCBM and other possible models. We first explore the effect that the fraction of annotated samples has on the results observed in Figure 4. Then, we investigate how the concept alignment threshold $\beta$ affects TabCBM's receptiveness to interventions. Finally, we include a comparison between interventions on TabCBMs, CBMs, CEMs, and Hybrid-CBMs showing how each of these models' performance varies as we intervene on more concepts.

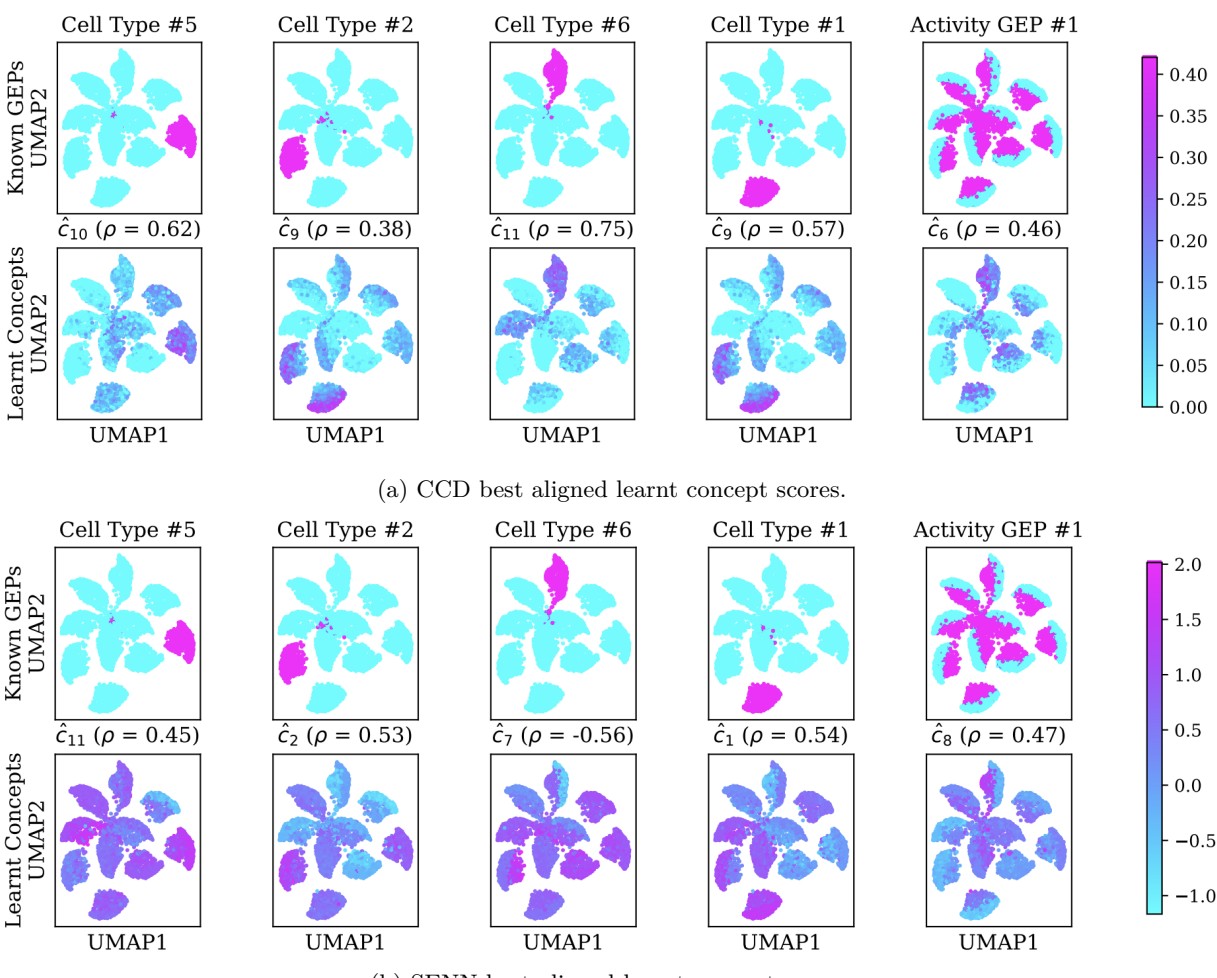

(a) CCD best aligned learnt concept scores.

(b) SENN best aligned learnt concept scores.

Figure 6: UMAP visualisations of the Synth-scRNA dataset showing the activation of a randomly selected selection of ground truth concepts (top row) and the activation of the learnt concept score by (a) CCD and (b) SENN with the highest absolute correlation with the respective ground truth concept (bottom row).

## H.1 Concept Interventions in Partially Concept-Supervised Setups

In Figures 7a and 7b, we show the test task accuracy when intervening on a TabCBM that was trained with $r = 25\%$ and $r = 50\%$ of the training samples provided with concept annotations. We notice that TabCBM appears to be quite robust to different levels of concept annotations, performing similarly to what we observe in the fully concept-supervised case in Figure 4 where all samples were provided with concept annotations. There seems to be a slight drop in accuracy when using only 25% of the concept annotations (see the right-most plot in the bottom row of Figure 7a), however, this does not represent a significant drop. More importantly, in all cases that we tested (as we varied the fraction of concept-annotated samples $r$ across $\{0.25, 0.5, 1\}$) we observe that TabCBM can discover unsupervised concepts which strongly align to concepts not provided during training and that, when intervened on, they can boost TabCBM's downstream task performance.

## H.2 Effect of Alignment Threshold

As discussed in Section 5.4, an important part of how we intervene on TabCBM's unsupervised concept scores involves building an alignment map $\alpha$ between these scores and known ground-truth concepts in the task of

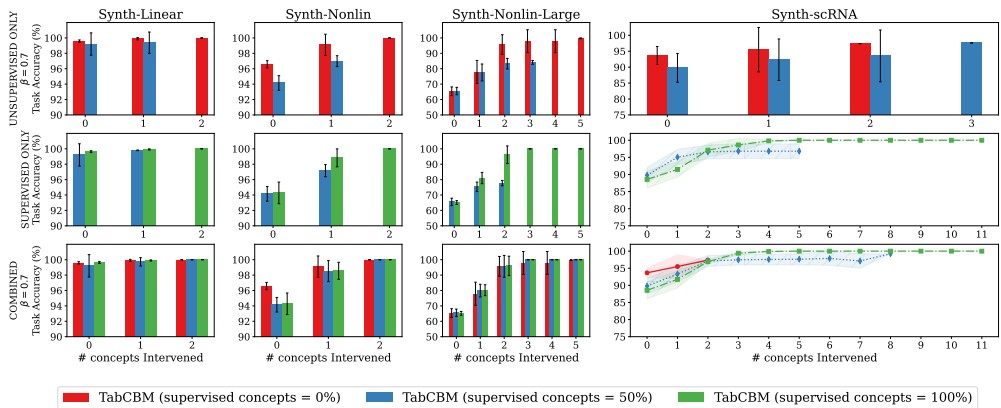

(a) Intervening when $r = 25\%$ of training samples for TabCBM are provided with concept annotations.

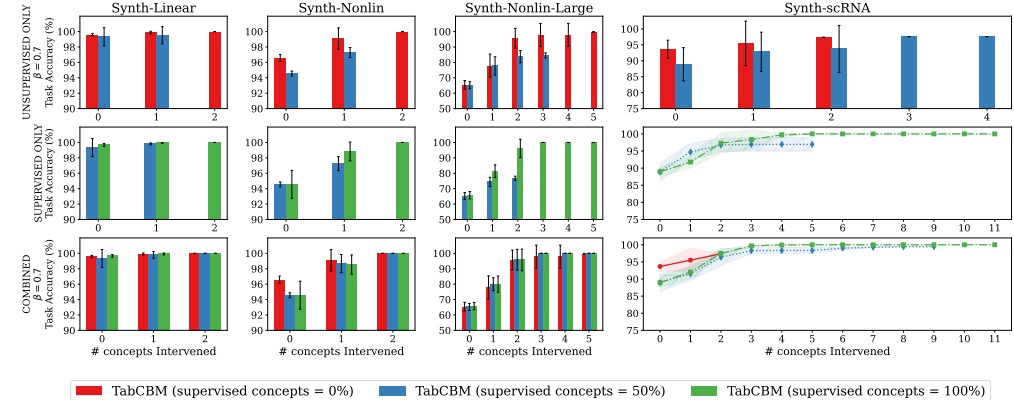

(b) Intervening when $r = 50\%$ of training samples for TabCBM are provided with concept annotations.

Figure 7: Task accuracy of TabCBMs when randomly intervening on a different number of concepts (x-axis) across multiple datasets (columns). Each row in each subfigure has the same semantics as its corresponding one in Figure 4.

interest. We argued in the same section that this map could be built by detecting strong linear correlations (which could be both negative or positive) and aligning discovered concept $\hat{c}_i$ with ground-truth concept $c_j$ if their empirical absolute linear correlation coefficients are greater than a threshold $\beta$. This threshold indicates how strong of a correlation should we look for to declare a discovered concept to be aligned with a ground-truth concept.

In our experiments reported in Figure 4, we fixed $\beta$ to $\beta = 0.7$, arguing that this indicates what we believe to be a strong enough correlation between two variables. Here, we show in Figures 8a and 8b how these results change if we set $\beta$ to 0.25 and 0.85, respectively. As one would expect, we observe that when $\beta$ is very low (i.e., $\beta = 0.25$) interventions in TabCBM can be damaging to TabCBM's performance in complex datasets such as *Synth-scRNA*. We attribute this to the fact that the constructed alignment map $\alpha$ may mark even slightly correlated concept scores and ground-truth concepts as aligned whose correlation can be attributed to the complexity and noise inherent to the dataset, rather than to a true semantic alignment. Nevertheless, we do not observe this in the case of the simpler synthetic datasets as TabCBM seems to be learning correlation scores whose absolute values are close to binary in nature (i.e., discovered concepts seem to be very strongly correlated to one and only one ground truth concept). In contrast, we observe that if $\beta$ is too large (e.g., $\beta = 0.85$), then we lose the ability to exploit slightly weaker correlations during intervention time (as we did when $\beta = 0.7$ in Figure 8), resulting in almost no discovered concepts identified as being aligned with ground-truth concepts in *Synth-scRNA*. This represents a cost as we know from Figure 4 that we could improve the model's test performance through interventions that arise from weaker correlations.

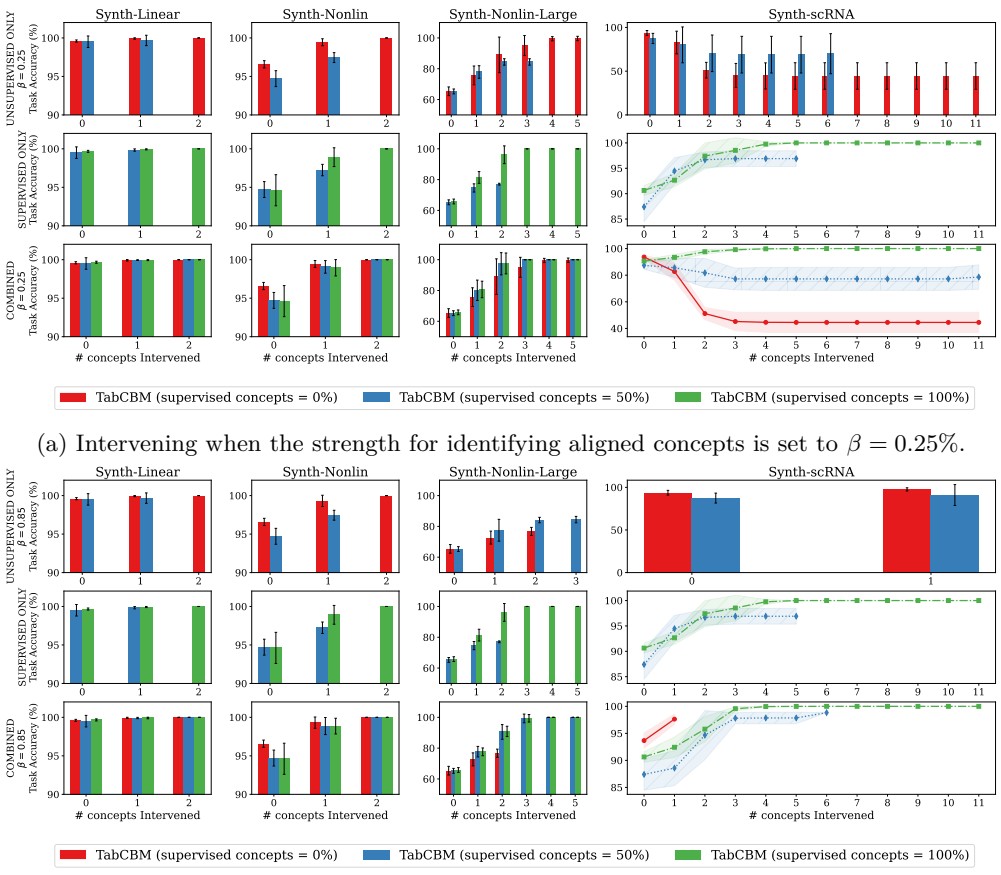

(a) Intervening when the strength for identifying aligned concepts is set to $\beta = 0.25\%$.

(b) Intervening when the strength for identifying aligned concepts is set to $\beta = 0.85\%$.

Figure 8: Task accuracy of TabCBMs when randomly intervening on a different number of concepts (x-axis) across multiple datasets (columns). Each row in each subfigure has the same semantics as its corresponding one in Figure 4.

In summary, these experiments suggest that one can think of $\beta$ as a parameter indicating how much *risk* we are willing to take when intervening with TabCBM. If it is very high, then we will only intervene on concepts that we are highly confident are aligned with ground-truth concepts, resulting in likely intervention gains as observed throughout our simpler synthetic datasets. On the other hand, if $\beta$ is very low, then we may incorrectly align a discovered concept with a ground truth concept which, when intervened on, may decrease the end performance of TabCBM due to misaligned semantics. Hence, we select $\beta = 0.7$ as a value that leverages this risk by taking advantage of possible strong correlations for improving test-time interventions, while avoiding misidentifying alignments that can lead to detrimental interventions.

### H.3 Intervention Comparison Across Concept-Supervised Methods

In Figure 9, we compare the effect that interventions have on the test accuracy across multiple concept-supervised methods (CBM, Hybrid-CBM, and CEM) for *Synth-Nonlin-Large* and *Synth-scRNA* as we vary the number of concepts provided during training between 50% of all concepts (top row) and 100% of all concepts (bottom row). We show our evaluation only on these datasets as, from our pool of concept-annotated datasets, they have enough concepts (i.e., 5 and 11 concepts, respectively) for us to study interesting behaviours emerging when we intervene on a varying number of concepts. As in Figure 4, in this study, we annotate all $r = 100\%$ of the training samples with the selected concept annotations and we use $\beta = 0.7$ when intervening on unsupervised concepts with TabCBM. The training setups for CEM, CBM, and Hybrid-CBM are the same as those used in Section 5.1. Notice that because TabCBM allows for interventions in unsu-

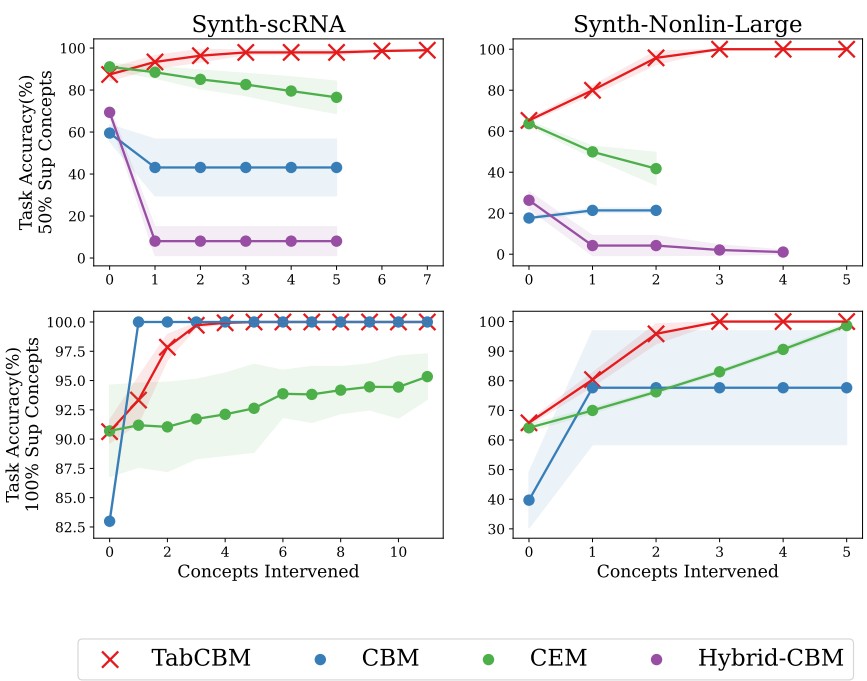

Figure 9: Task accuracy of different concept-supervised baselines in the *Synth-Nonlin-Large* and *Synth-scRNA* tasks as we increase the number of concepts intervened on a testing time. During training, we provide 50% of available concepts (top row) and 100% of available concepts (bottom row). For TabCBM, we include interventions both on supervised and unsupervised concepts.

pervised concepts, we can intervene on more concepts for this model than for the rest of the baselines (only applicable when the number of training concepts is less than the number of known ground-truth concepts in the dataset). Furthermore, we point out that because Hybrid-CBM is the same as a CBM when using all available concepts (there is no need for extra capacity in this case), we do not include this baseline in the results when intervening in models that were given training supervision for all concepts (bottom row figures).

Our results show that TabCBM can perform significantly better than the selected baselines in concept-incompleteness setup (i.e., when the number of training concepts is less than the number of ground truth concepts), as seen by the results displayed in Figure 9's top row. These results suggest that TabCBM can be an effective model to be used in real-world scenarios when it is likely that the set of concept annotations available at training time is incomplete with respect to the downstream task (Espinosa Zarlenga et al., 2022). When facing concept-complete setups, shown in Figure 9's bottom row where all required concepts for a task are given as training supervision, we see that TabCBM fares better or competitively against baselines. An exception for this can be found only in *Synth-scRNA*, where TabCBM falls behind CBMs when the number of interventions is small. We believe this to be the case because, given the mutual exclusiveness of *Synth-scRNA*'s identity GEP concepts, CBM can properly capture this property better than TabCBM as it does not have any extra constraints such as TabCBM's mask discovery. Nevertheless, we observe that as soon as a few concepts are intervened on, TabCBM's performance becomes similar to that of CBM in this dataset.

## I  Hyperparameter Ablations

In this section, we explore TabCBM's sensitivity to its hyperparameters by studying how they affect different metrics and properties of the resulting model. We first discuss how $k'$, the number of discovered concepts, affects TabCBM's accuracy and quality of learnt representations. Then, we explore how the different regularisation terms for TabCBM's loss affect its end performance. Finally, we conclude by giving some

recommendations on what are some good values for these regularisers that, from our experience, work well in practice without much fine-tuning.

## I.1 Effect of the Number of Discovered Concepts on TabCBM

Selecting the number of discovered concepts for TabCBM is one of the most important parts of constructing a model that is high-performing while maintaining a high level of interpretability. While setting this value can be strongly aided by domain-specific knowledge (e.g., setting $k' = k$ when we know the value of $k$ a priori), it is often the case and expectation that TabCBM will be trained on a task in which the true number of ground truth concepts is unknown. However, in Figure 10 we show that TabCBM's performance, is relatively robust to changes in its $k'$ parameter. We observe that only when it is severely capacity-constrained, as it is the case when $k' = 1$ in both *Synth-Linear* and *Synth-Nonlin*, do we observe a significant drop in its accuracy.

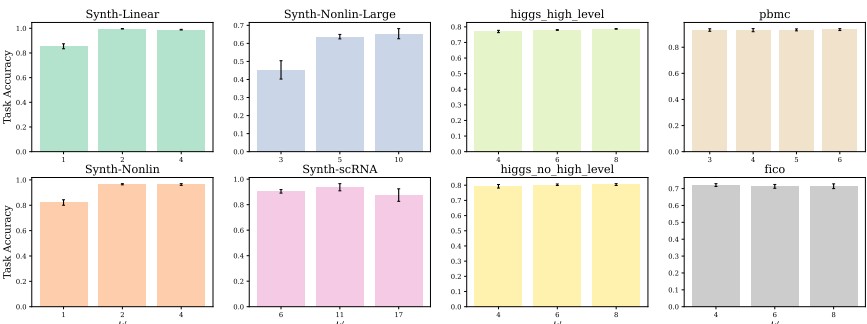

Figure 10: Test accuracies for TabCBM as we vary the number of discovered concepts $k'$ for all our tasks.

Although a similar behaviour as that observed with the task accuracy when we vary the value of $k'$ can be seen for the G-AUC metric, the same cannot be said of the M-AUC metric. We see this in Figure 11 where we observe that TabCBM can correctly identify all globally important features regardless of $k'$, provided it is not under-parameterised, resulting in a high G-AUC value for all variants evaluated. In contrast, we observe that the M-AUC metric seems to decrease as the value of $k'$ increases (particularly in the more complex synthetic datasets). We hypothesise that this is the case because in both *Synth-Nonlin-Large* and *Synth-scRNA* the ground-truth concept masks of some concepts are overlapping (e.g., a GEP's set of differentiated genes can overlap with the set of differentiated genes for another GEP). Therefore, when using a smaller $k'$, TabCBM's objective function can be minimised by fusing some of these concepts' masks and scores into a single concept. This leads to a higher M-AUC score than when $k'$ is larger because TabCBM does not need to correctly disentangle such fused concepts as it has to do when $k'$ is larger.

## I.2 Effect of TabCBM's Regularisers and General Recommendations

Besides $k'$, the next set of hyperparameters of importance for TabCBM is its different regulariser strengths $\{\lambda_{co}, \lambda_{div}, \lambda_{spec}\}$. These control how much we value concept coherence, concept diversity, and concept specificity, respectively, with respect to downstream task accuracy. In practice, when evaluating TabCBM we explored different combinations of these hyperparameters by varying them in the set $\{0.1, 1, 5, 10\}$. We observe that, although TabCBM's performance seems to be relatively robust to fluctuations in these parameters, the specificity strength is particularly important to fine-tune for TabCBM to correctly identify and discover concepts it was not given supervision for. To see this, in Figure 12 we show how the test task accuracy of TabCBM in the *Synth-Nonlin-Large* dataset changes as we modify all three regularisers while fixing the others to the parameters defined in Table 5 for this dataset. These results show that test accuracy fluctuates little for changes in both $\lambda_{co}$ and $\lambda_{div}$, but it is sensitive to changes in $\lambda_{spec}$. This is because there seems to be a threshold for $\lambda_{spec}$ at which TabCBM puts more of its attention into discovering masks first, and then uses these masks to simplify the problem of learning the downstream tasks. This helps in this particular example as by first finding meaningful concept masks, TabCBM eliminates a lot of noise and

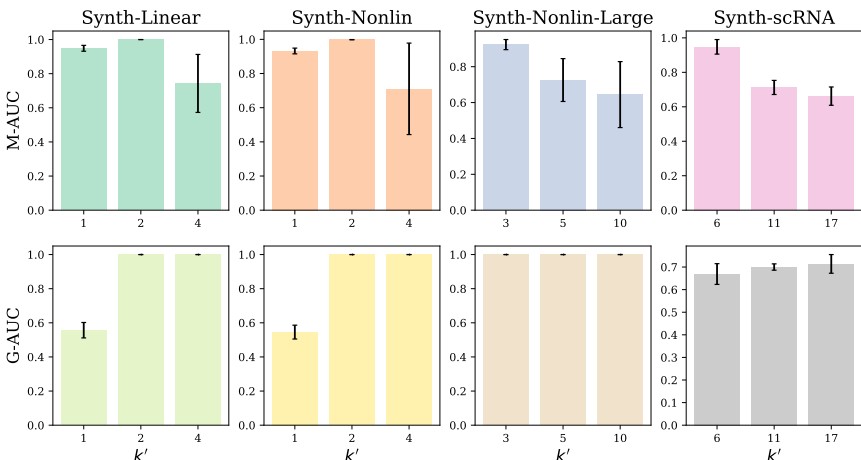

Figure 11: Test G-AUC and M-AUC scores for TabCBM as we vary the number of discovered concepts $k'$ for all our tasks that have concept annotations.

redundancy in the data in the same way that feature selection methods do; hence improving the learning dynamics after these masks have been discovered.

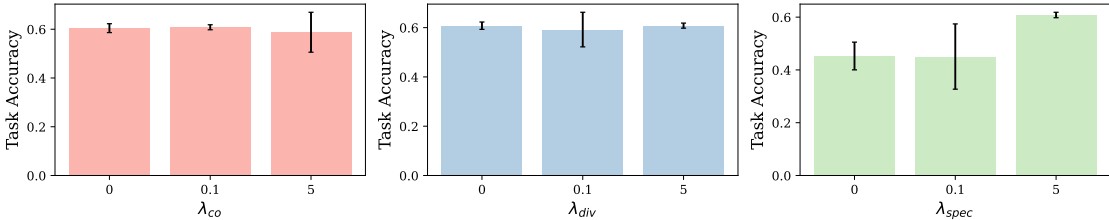

Figure 12: Test task accuracy of TabCBM in the *Synth-Nonlin-Large* dataset as we vary its loss regularisers $\{\lambda_{\mathrm{co}}, \lambda_{\mathrm{div}}, \lambda_{\mathrm{spec}}\}$ during training.

**Recommendations for loss hyperparameters** Our results above, together with our experience with TabCBM during evaluation, seem to suggest that the parameter that requires the most fine-tuning during training is $\lambda_{\mathrm{spec}}$. In practice, we observed that setting $\lambda_{\mathrm{co}} = 0.1$ while searching for values of $\lambda_{\mathrm{div}}$ and $\lambda_{\mathrm{spec}}$ in the set $\{0.1, 1, 5, 10\}$ under the constraint $\lambda_{\mathrm{div}} = \lambda_{\mathrm{spec}}$ yielded good results. If such a search is too expensive, then we found that setting $\lambda_{\mathrm{div}} = \lambda_{\mathrm{spec}} = 5$ resulted in good performance across all of our tasks without much fine-tuning.

## J   Differences Between Scoring Functions

As discussed in Section 4, in this work we opt to use as our concept scoring function $s^{(i)}$ the sigmoidal unnormalised inner product between $\phi(\tilde{\mathbf{x}}^{(i)})$ and $\rho^{(i)}(\tilde{\mathbf{x}}^{(i)})$. We chose this scoring function rather than other commonly used alternatives such as the cosine similarity (used in CCD (Yeh et al., 2020)) for two main reasons. First, using the cosine similarity for generating a concept score between 0 and 1 requires one to threshold the similarity so that its domain (i.e., [-1,1]) is clamped between 0 and 1 (as these vectors are not guaranteed to be non-negative). This is done in CCD by selecting all similarity scores below some hyperparameter $\beta$ to be clamped at 0. While this enables one to have magnitude-independent scores, it introduces the need for an extra hyperparameter and leads to a gradient-blocking operation, forbidding gradients to backpropagate to the vector-generating models $\phi$ and $\rho$ when their concept scores are clamped to zero. Second, as shown in Table 6 for Synth-Nonlin-Large, in practice we observe that using thresholded cosine similarities rather than our proposed activation function in TabCBMs can lead to drops in downstream

performance. We hypothesise that this is the case as our model can utilise the magnitude of these vectors to help it more easily express a concept's activation or deactivation after using a sigmoidal activation, something that normalised vectors may not be able to do. Because of these reasons, we leave the exploration of further scoring functions for future work and use the sigmoidal inner product in this paper.

Table 6: Differences in task accuracy and concept alignment scores (CAS) when using for TabCBM's concept scoring function $s^{(i)}$ (1) a $\beta$-thresholded cosine similarity, and (2) a sigmoidal dot product function. For simplicity, we show only results on the Synth-Nonlin-Large dataset and all TabCBMs are trained without any concept supervision by setting $k' = k$ and using the same hyperpameters as those described in Table 5.

| Method | Task Accuracy (%) | CAS (%) |
|---|---|---|
| TabCBM (sigmoidal inner product) | **62.78 $\pm$ 1.13** | **88.54 $\pm$ 4.49** |
| TabCBM (thresholded cosine similarity with $\beta = -0.5$) | 53.38 $\pm$ 2.01 | 74.73 $\pm$ 2.15 |
| TabCBM (thresholded cosine similarity with $\beta = 0$) | 53.48 $\pm$ 1.87 | 75.19 $\pm$ 1.96 |
| TabCBM (thresholded cosine similarity with $\beta = 0.5$) | 53.53 $\pm$ 1.47 | 74.87 $\pm$ 1.57 |

## K  Interpretability-accuracy Trade-off in TabCBMs

In our experiments exploring the task accuracy of concept-supervised TabCBMs, shown in Figure 2, we observe that TabCBM's task accuracy generally does not significantly vary when concept supervision is introduced. Nevertheless, we an exception is seen only in the Synth-scRNA task where a noticeable decrease in task accuracy is introduced as soon as concept supervision is provided to TabCBM. This suggests that an interpretability-accuracy trade-off may exist when concept supervision is introduced in complex tasks. In this section, we take a deeper look at this trade-off and show that, although it does indeed affect TabCBM in complex tasks, our models are still able to obtain competitive performance with respect to black-box baselines.

In Table 7 we summarise the performance of TabCBM across all synthetic tasks before and after receiving concept intervention. Our results suggest that *at its worst*, TabCBM drops around $\sim$7% in mean average performance in Synth-scRNA compared to an equivalent unsupervised version (i.e., a TabCBM trained without any concept supervision). Nevertheless, these differences are not necessarily significant as the variances are relatively large because of their high sensitivity to which concepts are provided with supervision. More importantly, when all concepts are provided supervision, we see a drop of only $\sim$3% in task accuracy. Such a drop is not significant when one considers that even with $\sim$3% less accuracy than its unsupervised version, TabCBM outperforms black-box models such as MLPs and falls behind other black-box models such as TabNet and XGboost by $\sim$0.5%. These negligible drops in performance, however, come with the increased benefit of TabCBM being able to generate faithful concept explanations for its predictions and being significantly more receptive to test-time interventions that can boost their performance way above that of black-box models (as seen in Figure 4). Finally, this trade-off may be corrected in practice by decreasing $\lambda_{\text{concept-sup}}$ during training, although this may lead to less accurate concept explanations.

Table 7: **Accuracy-interpretability trade-off**: effect of adding concept supervision to TabCBM across all synthetic datasets. We show the task accuracy (%) obtained as we change the number of supervised concepts given to the TabCBM during training. For reference, we include MLPs and TabNets as baselines to compare our model against.

| Method | TabCBM (0 Sup Concepts) | TabCBM (1 Sup Concepts) | TabCBM (50% Sup Concepts) | TabCBM (100% sup concepts) | MLP | TabNet |
|---|---|---|---|---|---|---|
| Linear | 99.6 $\pm$ 0.07 | 99.51 $\pm$ 0.37 | 99.51 $\pm$ 0.37 | **99.67 $\pm$ 0.07** | 97.57 $\pm$ 0.37 | 97.57 $\pm$ 0.37 |
| Synth-Nonlin | **96.58 $\pm$ 0.23** | 94.71 $\pm$ 0.52 | 94.71 $\pm$ 0.52 | 94.62 $\pm$ 1.01 | 87.65 $\pm$ 0.98 | 91.57 $\pm$ 0.48 |
| Synth-Nonlin-Large | **65.37 $\pm$ 1.41** | 65.35 $\pm$ 1.24 | 65.27 $\pm$ 0.79 | 65.81 $\pm$ 0.78 | 40.74 $\pm$ 6.42 | 51.01 $\pm$ 2.57 |
| scRNA | **93.66 $\pm$ 1.41** | 86.10 $\pm$ 2.53 | 87.40 $\pm$ 2.91 | 90.04 $\pm$ 1.78 | 73.87 $\pm$ 1.43 | 90.66 $\pm$ 1.10 |

## L  Details on representation learning metrics

In Table 8 we provide a brief summary of the metrics used in our experiments in Section 5.2.

Table 8: Summary of the metrics used as part of our evaluation in Section 5.2.

| Metric | Definition |
|---|---|
| Concept Alignment Score (CAS) (Espinosa Zarlenga et al., 2022) | This metric measures how strongly a learnt concept score $\hat{c}_i$ is aligned with a corresponding ground-truth concept $c_j$. It does this by clustering test samples based on $\hat{c}_i$ and measuring the homogeneity (Rosenberg & Hirschberg, 2007) (i.e., the *coherence*) of these clusters with respect to the ground-truth labels of $c_j$. To compute this metric for a set of learnt concepts, we first match each learnt concept score with the ground truth concept with the highest absolute Pearson correlation and then average the homogeneity scores produced for each matching. A high CAS score of 1 represents a perfect alignment between learnt concepts and ground-truth concepts. A low CAS score of 0 indicates perfect misalignment. |
| Mutual Information Gap (MIG) (Chen et al., 2018b) | This metric provides a quantitative measurement of how disentangled, or *diverse*, a set of learnt concepts is. It is computed by measuring how much more information about a ground-truth concept $c_i$ is encoded in the learnt concept $\hat{c}_a$ with the highest mutual information $I(\hat{c}_a; c_i)$ than in the learnt concept $\hat{c}_b$ with the second highest mutual information $I(\hat{c}_b; c_i)$. If the mean gap across all ground-truth concepts is large, then it implies that each ground-truth concept is being encoded by a single learnt concept. Otherwise, it implies that multiple learnt concepts are encoding redundant information about a ground truth concept. |
| $R^4$ (Ross & Doshi-Velez, 2021) | The $R^4$ metric measures whether there exists a *bijective alignment* function between learnt concept scores and ground-truth concept labels. Therefore, it attempts to capture whether learnt concepts are coherent and diverse enough to completely capture all ground-truth concepts. This metric operates by training two nonlinear univariate models for each (learnt-concept, ground-truth concept)-pair, one model for each direction, and using their coefficients of determination $R^2$ to quantify how closely can these two variables be represented using a bijective mapping. A final score is then produced by averaging the maximum $R^2$ values obtained for all ground-truth concepts. A high $R^4$ score indicates the existence of a bijection between learnt concept scores and ground-truth concepts. |
| DCI Disentanglement (Eastwood & Williams, 2018a) | This metric captures concept *diversity* by measuring whether each learnt concept is aligned with at most one ground truth concept. It is computed by averaging the complement entropies of the probabilities that a learnt concept $\hat{c}_i$ is rendered "important" when predicting ground-truth concept $c_j$. The importance scores used to construct such a probability distribution are computed using a simple linear regression model between every learnt concept and every ground-truth concept. A high DCI disentanglement indicates that all learnt concepts capture one and only one ground-truth concept. |
| DCI Completeness (Eastwood & Williams, 2018a) | When the set of ground-truth concepts is fully descriptive of the downstream task, this metric captures *concept completeness* by measuring the degree to which each ground-truth concept is captured by at least one learnt concept. It estimates this by computing the complement entropy of the probability of learnt concept $\hat{c}_i$ being the only "important" learnt concept to predict ground-truth concept $\hat{c}_i$ (with importance computed as above). A high DCI completeness indicates that all ground-truth concepts are captured by one and only one learnt concept. |
| DCI Informativeness (Eastwood & Williams, 2018a) | This metric captures another aspect of *concept completeness* by looking at how predictive the *overall* set of learnt concept scores is of each known ground-truth concept. This is measured using the average prediction error obtained when training a simple classifier to predict each ground-truth concept from $\hat{\mathbf{c}}$. A high DCI informativeness indicates that all ground-truth concepts can be accurately predicted using the set of learnt concept scores. |

## M   Code Used and Hardware Infrastructure

For the experiments reported in this paper, we built a code base on top of the code made available by Espinosa Zarlenga et al. (2023) and by Kazhdan et al. (2021) (under an MIT and Apache 2.0 licenses, respectively) to train the CCD, SENN, and CBM baselines. For both the CEM and Hybrid-CEM baselines, we used the official implementation by Espinosa Zarlenga et al. (2022). For both XGBoost and LightGBM, we made use of the official implementations of both frameworks. For TabNet we extended the open-sourced MIT-licensed implementation by DreamQuark AI[3]. Similarly, for TabTransformer we extended the open sourced MIT-licensed implementation by Phil Wang[4].

We built our code base using a combination of TensorFlow (Abadi et al., 2016) and PyTorch (Paszke et al., 2019) and implemented TabCBM in TensorFlow. All of the code needed to reproduce our results, and use our model through a simple API, has been made public at https://github.com/mateoespinosa/tabcbm via an MIT license.

---

[3]https://github.com/dreamquark-ai/tabnet
[4]https://github.com/lucidrains

