# OpenReview forum: "TabCBM: Concept-based Interpretable Neural Networks for Tabular Data"
_TMLR — Accepted by TMLR_

### Review · Reviewer_NmM7 · 2023-04-30

**Summary Of Contributions:**

The present paper applies concept bottleneck models to tabular data. To this end, first, the authors define what concepts are in the domain of tabular data.
Which is the interaction between relevant features, including their activation. The proposed method includes, next to the original concept bottleneck, which was originally introduced in the vision domain, a concept generator to not only rely on supervised concept signals but also provide see ability to unsupervised train such a network. The proposed methodology is evaluated on synthetic as well as real-world tabular data, also highlighting human interventions, which are, next to interpretability, an essential motivation of concept bottleneck models.

**Audience:**

Yes

**Claims And Evidence:**

Yes

**Requested Changes:**

In general, the present paper seems to be ready for acceptance. The presentation of Fig. 2 can be improved by increasing the font size of the plots' labels. Further,  in some cases, TabCBM seems to outperform other approaches by a large margin (Synth-Nonlin). Can you further elaborate on that?



**Strengths And Weaknesses:**

**Strength:**
- The paper addresses the important topic of interpretable deep models and, even more importantly, networks that allow for human interventions.
- The paper is very well written and easy to follow, especially the description of the original CBMs, the definition of tabular concepts, and the introduction of TabCBM.
- Experiments seem to be well conducted, and TabCBMs are exhaustively on synthetic and real-world data, including ablations and the application of interventions.
- The paper is well describing the limitations of future work.

**Weaknesses:**
- Limited novelty since the contribution is mainly the application of CBMs to tabular data and tasks. However, the paper provides important insights. Therefore this weakness is negotiable.

---

> ### Author Response · Authors · 2023-05-28
> **Reply to review**
>
> Dear Reviewer NmM7,
>
> Thank you for your invaluable feedback - we are glad that you found our paper easy to follow and our evaluation well-conducted. Below we elaborate on your concern related to the novelty of this work. We also provide an answer to your question regarding TabCBM’s performance boost over existing baselines in Synth-Nonlin. As for your comment regarding the size of Figure 2, we agree that we could have done a better job in the font and plot sizes and we have updated our manuscript to include this.
>
> ### Novelty with respect to CBMs
>
> While we agree that the general idea behind Concept Bottleneck Models is a key building block for our method, we want to emphasise that our proposed approach distinguishes itself from CBMs in the following key aspects: (1) CBMs cannot discover new concepts in an unsupervised manner. This is key as even when one introduces extra unsupervised capacity in a CBM’s bottleneck, it has been shown that these unsupervised activations do not end up aligning with extra unseen concepts even for simple synthetic tasks [1]. In contrast, in our work we are able to learn and discover concepts that align with unseen ground-truth concepts without any supervision. (2) Our method is able to discover the set of features used to construct each concept, allowing one to understand the features composing each concept and possibly assign semantics to the concept if this is a discovered concept. In contrast, CBMs do not offer such a mechanism and have in fact been shown to lead to very noisy feature attribute maps generated via traditional saliency methods for DNNs [2]. (3) Our work is the first work to discuss what a concept may entail in a tabular domain, a very important domain for safety-critical tasks (e.g., medicine) and something that, to the best of our knowledge, has not been previously explored by previous work.
>
>
> ### TabCBM’s outperforming baselines significantly in Synth-Nonlin
>
> This is a really good question which we hope to further explore in future work. For the time being our hypothesis is that TabCBM significantly outperforms other methods in this task due to it being able to learn to select **multiple subsets of features** from which distinct binary functions can provide important task-related information. This process enables TabCBM to get rid of a lot of the noise in the task’s feature space (90% of all features are irrelevant), therefore avoiding overfitting irrelevant features that other methods may overfit (e.g., MLPs). This may also partially explain why when we increase the feature space, as in Synth-Nonlin-Large, we see even a larger gap. We note that this hypothesis is motivated by results provided by recent work [3] that show how using an ensemble of feature selection masks can perform better than using a single or no feature selection mask (we discuss this work in our discussion). Nevertheless, we reiterate that different from such methods, our training objective does not constrain each learnt mask to select different features (as group feature selection does) and we aim to discover feature groups that map to tabular concepts. This is particularly important for tasks such as Synth-Nonlin-Large where the subsets of features used for ground-truth concept overlap.
>
> ### References
>
> [1] Mahinpei, Anita, et al. "Promises and pitfalls of black-box concept learning models." arXiv preprint arXiv:2106.13314 (2021).
>
> [2] Margeloiu, Andrei, et al. "Do concept bottleneck models learn as intended?." arXiv preprint arXiv:2105.04289 (2021).
>
> [3] Imrie, Fergus, et al. "Composite Feature Selection using Deep Ensembles." NeurIPS 2022.

---

### Review · Reviewer_ez7U · 2023-05-01

**Summary Of Contributions:**

This work introduces TabCBM, a Concept Bottleneck Model for tabular data, which
combines a feature selection approach based on (Lee et al., 2021) to learn a
concept-level feature mask, with a concept embedding approach based on (Yeh et
al., 2020).
Specifically, TabCBM produces a set of concepts that consist of a feature mask
as well as a concept score function.
A motivation for concepts in tabular data is provided, as well as a set
of desiderata for meaningful concepts, which is subsequently used to motivate
the training objective of TabCBM, as well as the evaluation metrics in the
following experiments.
The performance of TabCBM without concept supervision (the a-priori knowledge
of concepts) is empirically compared to 7 other models on 4 toy datasets and 3
real-world datasets, as well as 3 other models with concept supervision on the
same 4 toy datasets, on which TabCBM shows equal to superior accuracy on all
datasets.
The alignment of the identified concepts of TabCBM is empirically shown as
superior compared to 3 baselines, including PCA, using 6 metrics from the
literature, related to the declared concept desiderata. Additionally, the
alignment is qualitatively compared on one dataset, showing good alignment of a
few concepts, which is shown to not be the case for two other baseline concept
learning methods from the literature.
It is empirically shown that TabCBM's feature mask provides a global feature
importance, using 4 toy-dataset with known ground-truth masks.
The work also provides an empirical evaluation on the intervention of concepts
on a toy dataset, showing how the correction of a few concepts, regardless of
whether they were known a-priori, can increase the model accuracy of TabCBM.
The work concludes with conceptionally relating TabCBM to group feature
selection, as well as discussing limitations of TabCBM, specifically the
difficulty of the selection of the number of concepts, computational growth
with a high number of concepts or the dimensionality of concept embeddings, and
its architectural complexity and high number of hyper-parameters.


**Audience:**

Yes

**Broader Impact Concerns:**

None.

**Claims And Evidence:**

Yes

**Requested Changes:**

### Method

#### Major (non-critical)

- Address the weaknesses above about the concept score.

- Explain the training of $\rho$ and $\phi$ as pointed out above.

- Relate the desiderata to other work.

#### Minor (non-critical)

- Desiderata 1. Should be pointed out to be task specific, and it should relate
  to the task in Eq. (1)
- Desiderata 3. It should be pointed out here that the similarity is probably
  with respect to some latent similarity.

- $\Psi$ is defined as the t-nearest-neighbors, but the metric is only implied to
  be L2. This should be added.

- In Eq. (1), the inner product should be normalized, otherwise this term may
  only reduce the amplitude of the concepts, same for Eq. (2). This should be
  either discussed, or changed.

### Experiments

#### Critical

- Address the lack of realistic data and the alignment therein of unsupervised
  TabCBM concepts with ground-truth concepts, or explicitly disregard the
  interpretability claim in real-world settings without concept annotations.

- Address the interpretability trade-off with supervised concepts.

#### Minor (non-critical)

- The number of trials for the experiments should be provided.

- With respect to the performance, in the end of 5.1 paragraph "Concept
  unsupervised" it is stated that *"even without concept supervision, TabCBM
  does not sacrifice the performance seen in state-of-the-art black-box models"*.
  My intuition would be that the trade-off is stronger *with* supervision, as
  indicated by Fig. 2a, thus, this statement is misleading.

- Fig. 3 could use some more explanation, especially for Cell Type 5/$\hat
  c_9$, which has its sign flipped.


### Miscellaneous (non-critical)

- In the Eq. below Fig. 4, maybe pre-define the argmax index to improve readability

- In the Eq. below Fig. 4, maybe add directly after the percentiles that they
  represent the positive/negative activations of the concept within the
  training set

- Fig. 2 is a little bit too small.

- It may be beneficial to label all equations to make them easier to reference by readers

- It would not hurt to write down $\mathcal{L}_\text{concept-sup}$
  explicitly instead of describing it.

- Use a full-line equation for the definition of $s^{(i)}$

- The scatter plots take a long time to load, consider using a raster graphic format, e.g. PNG

- Fig. 4 final sentence: ...regardless of whether the[ir -> y] received training...


**Strengths And Weaknesses:**

### Strengths
The paper is overall in a very good shape.

- Concepts in tabular data are well motivated on previous work.
  Thus, TabCBM is well motivated, relating to and building on previous work on
  feature selection and concept embedding.

- The desiderata of concepts in tabular data motivate the objective and
  follow-up methods well.

- The research questions in the beginning of the extensive experiments section
  provides a good overview.

- For all experiments, there are at least from two (CCD and SENN in the
  unsupervised case), to 3 (CBM, CEM, Hybrid-CBM in the supervised case)
  up-to-date baselines for which TabCBM is compared to.

- The supplementary is very extensive and answers many questions, providing a
  good foundation for high reproducibility.


### Weaknesses

### Method

- The concept score is the sigmoid of the inner product of a concept vector and
  a latent encoding, as opposed to CCD where the score is the cosine
  similarity. Using the sigmoid of the inner product without normalization has
  two weaknesses:
  - The amplitude of the latent code/ concept vector influence the
    activation stronger then the respective direction.
  - Using a sigmoid activation on the inner product will result in a score
    of 0.5 when the directions are orthogonal and approach 0 when they are
    parallel with a different sign.

- The training of $\rho$ and $\phi$ lacks some explanation. Are they trained in
  parallel with the same concept objective?

- The desiderata, while providing a good motivation, would be better to relate
  to previous work.

### Experiments

- With the exception of the scRNA dataset, all concept related experiments are
  only conducted on synthetic data. This does not provide a lot of information
  on how useful TabCBM's concepts are on real world data.
  It is hard to say how much semantic meaning is carried by the unsupervised
  concepts. This is somewhat addressed by the intervention experiment, where it
  is also discussed that there are only few concepts in scRNA aligned with
  ground truth concepts. This is in important difference to the synthetic data,
  and should be at least discussed in greater detail. It would be much better
  if the experiments would include another realistic dataset with ground-truth
  concepts, although I understand this may be hard to come by.

- The interpretability trade-off of TabCBM is not discussed. TabCBM loses
  accuracy when using supervised concepts, i.e. in scRNA Fig. 2a. How does it
  perform in its worst case with supervised concepts when compared to
  (non-interpretable) models?

---

> ### Author Response · Authors · 2023-05-28
> **Reply to review 1/2**
>
> Dear Reviewer ez7U,
>
> We are grateful for all your detailed feedback and suggestions. We have done our best to take into account all the suggested revisions (marked in red in our updated manuscript) and we answer specific concerns below:
>
> ### Use of unnormalised inner-product rather than cosine similarity in our loss terms
>
> Thank you so much for catching this. This is a typo in our manuscript and, as you rightly discussed, we do in fact use the cosine similarity rather than the unnormalised inner products for computing both $\mathcal{L}_\text{co}$ and $\mathcal{L}_\text{div}$ (now equations (3) and (4), respectively). This can be seen in lines 525-559 of `models/tabcbm.py` in our submitted TabCBM code in the supplementary material. We apologise for this typo and we have updated the loss functions accordingly to indicate that we use the cosine similarity rather than the plain inner product.
>
> ### Sigmoidal inner product for concept score rather than normalised inner product (i.e., cosine similarity)
>
> This is a great question. In this work, we opt to use as our concept scoring function $s^{(i)}$ the sigmoidal unnormalised inner product between $\phi(\mathbf{\tilde{x}}^{(i)})$ and $\rho^{(i)}(\mathbf{\tilde{x}}^{(i)})$ rather than their cosine similarity for two main reasons. First, using the cosine similarity for generating a concept score between 0 and 1 requires one to threshold the similarity so that its domain (i.e., [-1,1]) is clamped between 0 and 1 (as these vectors are not guaranteed to be non-negative). This is done in CCD by selecting all similarity scores below some hyperparameter $\beta$ to be clamped at 0. While this enables one to have magnitude-independent scores, it introduces the need for an extra hyperparameter and leads to a gradient-blocking operation, forbidding gradients to backpropagate to the vector-generating models when their concept scores are clamped to zero. Second, in practice, we observe that using cosine similarities rather than our proposed activation function in TabCBMs can lead to drops in downstream performance. We include an example of such an instance for the Synth-Nonlin-Large task in a new Appendix (App. J). We hypothesise that this is the case as our model can utilise the magnitude of these vectors to help it more easily express a concept's activation or deactivation after using a sigmoidal activation. To clarify our rationale for our choice of scoring function for future readers, we have updated our manuscript in Section 4 (when introducing $s^{(i)}$)  so that it summarises the points made in this reply.
>
> ### Training of $\rho$ an $\phi$
>
> We train both $\rho$ and $\phi$ in an end-to-end fashion in conjunction with all the other pieces of TabCBM. We realise that the use of the expression “in parallel” was very misleading on our end and have addressed this usage in Section 4 to explicitly indicate how these components are trained.
>
> ### Lack of other real-world datasets and tasks in our alignment evaluation
>
> This is a limitation we openly share with the reviewer. Nevertheless, we want to highlight that because we are the first ones to propose and discuss the idea of extracting high-level concepts in tabular tasks, finding non-synthetic datasets with such annotations has proven to be extremely difficult. Currently, we can construct “semi-synthetic” datasets from real-world tabular tasks where the label is synthetically constructed so that the underlying concepts are known at test-time (as in other previous works e.g., [1]). However, it has proven difficult to find any real-world datasets that have these sorts of concept annotations in them **and** whose real-world tasks are also a function of these concepts. Therefore, in this work we decided to focus more on the methodology and on the proposed novel definition of a tabular concept from which we show that a concept-based interpretable model for tabular data can be designed. As part of our intended research plan after this work, however, we are hoping to work with domain experts in biology to explore how this method can be used to discover novel insights on real-world tasks.
>
> To better address the possibility of misrepresenting our alignment results, we clarified why we did not use other real-world datasets for our alignment experiments in our Discussion section and listed it as a possible venue for future work. Furthermore, we added a statement indicating the limitation of our alignment results in the last paragraph of Section 5.4 and we softened our real-world alignment claims in the abstract to reflect this.

---

> > ### Author Response · Authors · 2023-05-28
> > **Reply to review 2/2**
> >
> > ### Interpretability-accuracy trade-off for TabCBM
> >
> > Thank you for pointing this out. In our task-performance experiments for concept-supervised training of TabCBMs, summarised in Figure 2a, we observe that with the exception of scRNA, TabCBM’s task accuracy does not significantly decrease when concept supervision is introduced. This suggests that, as mentioned by the reviewer, an interpretability-accuracy trade-off may be present when concept supervision is introduced in complex tasks, although this trade-off is not a significant one (as seen in, say, Hybrid-CBMs). Following this suggestion, we have updated the last paragraph of Section 5.1 to discuss this trade-off and included an Appendix (App. K) where we discuss this result more clearly with the help of a table that better represents the results visually displayed in Figure 2.
> >
> > Our results suggest that at its worst, TabCBM drops around $\sim$7% in mean average performance in Synth-scRNA compared to an equivalent unsupervised version (i.e., a TabCBM trained without any concept supervision). Nevertheless, these differences are not necessarily significant as the variances are relatively large because of their high sensitivity to which concepts are provided with supervision. More importantly, when all concepts are provided supervision, we see a drop of only $\sim$3% in task accuracy compared to unsupervised TabCBMs. Such a drop is not significant when one considers that even with $\sim$3\% less accuracy than its unsupervised version, TabCBM significantly outperforms black-box models such as MLPs and falls behind other black-box models such as TabNet and XGboost by $\sim$0.5\%. These negligible drops in performance, however, come with the increased benefit of TabCBM being able to generate faithful concept explanations for its predictions and being significantly more receptive to test-time interventions that can boost their performance way above that of black-box models (as seen in Figure 4). Finally, this trade-off may be corrected in practice by decreasing $\lambda_\text{concept-sup}$ during training, although this may lead to less accurate concept explanations.
> >
> >
> > ### Possibly misleading sentence at the end of 5.1
> >
> > We can now see how this can be potentially misleading and have adjusted the language used in that paragraph so that it now says: “...our aggregate results strongly indicate that TabCBM does not sacrifice the performance seen in state-of-the-art black-box models when it is not provided with supervision.” We hope that this rewording makes it clear that we intended to say that when no supervision is provided, our method is competitive against state-of-the-art black-box models.
> >
> > ### Relating desiderata to previous related work and clarifying some of its terms
> >
> > As suggested, we have indicated the relationship between desideratum 1 (“concept completeness”) and a specific downstream task whose loss is measured by  $L_\text{task}$. Similarly, we have clarified some of the details of desideratum 3 (coherence) to be more explicit on how the neighbours are computed. Finally,  we have connected some of our desiderata with existing previous work when possible. You can see all of these updates in our manuscript by going over the red text in Section 4.
> >
> > ### Number of trials for our experiments
> >
> > As discussed in Appendix E, all of our experiments are done by averaging each metric across five different random seeds for each model. To make this clearer, we have included a line detailing the number of trials at the beginning of Section 5.
> >
> > ### Why is one of the signs flipped for the correlations shown in Figure 3?
> >
> > Notice that although TabCBM is able to discover some of the underlying concepts for a given task, it has no constraints regarding how it should label a given concept’s activation or deactivation. The only constraint it has is that it must use `0` and `1` to indicate opposite states of a concept. Therefore, it is perfectly allowed to use `0` to indicate that concept is active even though in our ground concept annotations the corresponding concept is annotated with `1`s indicating its activation. This implies that when assigning semantics to a concept discovered by TabCBM, one must consider that the labels it uses for a given concept may be “flipped”. This is why we use the maximum absolute correlation value, rather than simply the maximum correlation when plotting Figure 4. Concept $\hat{c}_9$ shows a case in which TabCBM indeed learnt the complement label for a ground-truth concept. We have added a short comment in Figure 3 to further clarify why we consider negative correlations.
> >
> >
> > ### Miscellaneous non-critical suggestions
> >
> > We thank you for all these minor suggestions. They are all very helpful and have been incorporated into our updated version.
> >
> > ### References
> >
> > [1] Yeh, Chih-Kuan, et al. "On completeness-aware concept-based explanations in deep neural networks." Advances in Neural Information Processing Systems 33 (2020): 20554-20565.

---

> > > ### Comment · Reviewer_ez7U · 2023-06-16
> > > **Minor Points**
> > >
> > > Thank you for addressing all of my concerns.
> > > I am satisfied and feel the paper is in very good shape.
> > >
> > > I only noticed a few very minor points while going through the revision:
> > >
> > > - virtually none of the formulas that end a sentence feature a period at the end of the line
> > > - the number of trials were not reported as indicated by the author's reply, however, I initially overlooked them in Appendix E, which I think is sufficient
> > > - Figures 3 and 5 now load much faster after using a raster graphics format, however, the resolution is a little too low, and their sizes should be optimized a little bit better to match the surrounding text size

---

### Review · Reviewer_mG3r · 2023-05-07

**Summary Of Contributions:**


In this paper, a new type of neural network, TabCMB, is proposed to train networks that capture “concepts” in tabular data and is able to ground its predictions on these “concepts”. The major contributions include the architecture of TabCMB itself, ways to train TabCMB and empirical studies showing TabCMB outperforms baseline works on the dimension of  “interpretability” for the task at hand.


**Audience:**

Yes

**Claims And Evidence:**

Yes

**Requested Changes:**

My recommendations are actually included in the weakness section. In summary,

1) I want to see a concrete example of "meta-concept" found by the TabCBM network, what does that mean from domain experts' perspective and is that more interpretable to them. Maybe defining "more interpretable" is helpful to convince the audience this model is indeed more interpretable than just throwing things into a tabular transformer.

2) The presentation of the empirical study should be improved to have focuses, clear definitions and metrics, and what to expect from the results.



**Strengths And Weaknesses:**

### Strength

A rich amount of empirical studies is the greatest strength for this work. I appreciate the authors’ efforts on quantitative evaluation for the proposed work over the baselines. Results in Section 5 have answered the 4 research questions introduced in the beginning. Especially, I find the results in concept score alignment interesting and I am surprised to see the model learns aligned scores in Figure 3 without supervision.


### Weakness

I have a few concerns for motivation, the training and the presentation of the current manuscript.

First of all, I have to admit I am not convinced that having this definition of “tabular concept” helps us to build models that are more interpretable. From my reading, this definition of (tabular) “concept” means some groups of input features are “concepts”, justified in one dataset where some features together are regarded as “delinquency” and can be used without other features to predict the default. Concept bottleneck model is a good fit for image data as pixels are not the unit for humans – we do not see pixels and we see hands, wings, etc. However, most tabular data has features with very concrete meanings, e.g. credit scores, income, heights. As an engineer with a computer science background, they are already “concepts” to me. However, grouping these features, without introducing domain knowledge in detail, are not creating a more interpretable “concept” to me. Transferring CBM ideas to tabular needs to answer two questions that: 1) given most tabular input features are already semantically meaningful enough and why one needs some “meta-concepts”; and 2) do these “meta-concepts” make sense to domain experts.

Secondly, this training objective is very complex and I have no idea how many efforts one needs to input to do parameter turning. The authors are very ambitious to include all properties of the learned concept vectors into the loss. As a result, every property becomes a regularization term. Moreover, there is another hyper-parameter to balance between concept supervision and unsupervised learning. My experience in training both small and large models tells me that without that amount of hyper-parameters, it requires so much effort in tuning and sometimes the result is just an overfitting to some set of hyper-parameters. I think hyper-parameter searching processes and bags of training tricks need to be presented (maybe it is in the appendix already and please forgive me if I have missed the pointers). If you ask me for advice on this, I would decrease the number of regularizations and think about what are fundamentally useful regularizations.


Lastly, the paper is too dense in the empirical part (and probably in a bad way). The authors are ambitious (which is good!) to answer 4 research questions that can quantitatively the proposed method. However, many metrics are just citing references and do not define them in the main paper. Moreover, I am lost many times and asking myself why they are evaluating this and what I should expect when looking at these tables. In addition, I do not understand why these metrics show that TabCMB is more interpretable if we do not have ground-truth “meta-concepts” at hand, which I assume is often the case. Can you evaluate mostly on real datasets and provide more examples & visualizations to explain what are learned “meta-concepts”?

---

> ### Author Response · Authors · 2023-05-28
> **Reply to review 1/2**
>
> Dear Reviewer mG3r,
>
> Thank you for your very insightful feedback. We are glad to read that you found our evaluation rich and interesting. We use this opportunity to address your concerns and questions and explain how we have updated our manuscript accordingly. To make new changes in our manuscript easier to track, we have marked them in red in our updated submission. Please let us know if anything is unclear after this and/or if we are able to further clarify or address some of your concerns with our work.
>
>
> ### What is a concrete “meta-concept” in TabCBM, why are they useful, and how can a domain expert detect it?
>
> While we agree with the reviewer that it is common for input features in tabular data to be  “high-level concepts” themselves, the generality of this phenomenon is remarkably domain-dependent. For example, in life sciences in general, and in genomics in particular, it is not the case that input features are high-level concepts. We specifically focus on such domains for this reason. In particular, in our paper we discuss single-cell RNA data (see first paragraph of Section 3 as well as the description of Synth-scRNA in Section 5), where one has in the order of hundreds or thousands of gene expressions for each cell. In this domain, a specific gene’s expression is rarely indicative of a global behaviour or pattern given the high involvement of multiple genes in most cell biological processes. Instead, one is commonly interested in the co-regulation of multiple genes through what is referred to as gene expression programs (GEPs) (see [1] for a comprehensive description of what these programs entail). Such programs can be formed by groups of genes whose expression, or lack thereof, can be used to understand whether a biological process is undergoing in the cell. For example, there are several known GEPs that are associated with different stages of a cell’s cycle or with specific cell identities (also referred to as identity GEPs). A tabular concept can therefore be used to represent, say, a hypoxia GEP by selecting genes known to be involved in hypoxia (e.g., VEGFA, PGK1, CA9, etc [1]) and having as its activation function a binary indicator that is on when these genes are **jointly** highly expressed and the cell is undergoing hypoxia. Through such gene expression programs, biologists can abstract the nuances and complexities involved with specific gene expressions (i.e., features in these tasks) and instead reason about a specific task of interest in terms of biological processes which they as experts are familiar with.
>
> In this example, an expert may detect or interpret a TabCBM concept as being aligned with a known GEP in practice by using a handful of samples they are familiar with or using a limited subset of samples with ground-truth GEP annotations in them. With such a dataset at hand, an expert can explore if any of TabCBM’s concept scores is highly correlated with any of the GEPs they are aware of. This is not very different from how current genomic workloads work when, say, experts use topic models to identify cell types within a population (although traditional topic models do not provide masks of selected genes for each topic, making them less powerful than concepts discovered by TabCBM). The fact that TabCBM generates feature masks for each of its concerts also enables experts to confirm the suspected semantics of a given concept by looking at the genes it selected and making sure that they fit with what they would expect for the GEP that the concept may be aligned with. Once an expert has identified the discovered concepts, the expert can (1) use this concept to understand how TabCBM is reasoning about the task of interest to learn novel insights about that task and (2) intervene on this concept to improve the overall method’s performance.

---

> > ### Author Response · Authors · 2023-05-28
> > **Reply to review 2/2**
> >
> > ### Evaluation with more real-world datasets
> >
> > Given that we are the first ones to propose and discuss the idea of extracting high-level concepts in tabular tasks, finding non-synthetic datasets with such annotations has proven to be extremely difficult. This has limited our ability to evaluate our dataset on other real-world datasets than the ones we include in the paper, as analysis would require close interaction with domain experts, which is out of the scope of this work. Therefore, in this work, we decided to focus more on the methodology and definition of a tabular concept, showing that we can design an architecture that is able to extract such concepts in synthetic domains while maintaining a competitive (or better) accuracy than state-of-the-art methods on both synthetic and real-world tasks. We want to reiterate the latter benefit as even if one is not interested in the interpretability part of our architecture, our results show that TabCBMs outperform tabular transformers and TabNets in both our synthetic and real-world tasks.
> >
> >
> > For clarity, we have now modified our Discussion section, as well as some of our claims throughout the paper, to explicitly explain why we did not include further analysis in real-world datasets and that this is exactly what we intend to explore next as part of our follow-up work. Changes are shown in our updated manuscript in red.
> >
> >
> > ### Complexity of objective function and hyperparameter tuning
> >
> > We agree with the reviewer that the number of hyperparameters involved in its loss function is a limitation. In fact, we discussed this in Section 6 while addressing some concerns about the sensitivity of the loss function to these hyperparameters in Appendix I.2. Our results show that our method is robust to small changes to $\lambda_\text{co}$ and $\lambda_\text{div}$ while being mostly affected by $\lambda_\text{spec}$. In that same appendix, we further introduced some general recommendations on how to train these models by simply focusing on varying $\lambda\text{spec}$ while keeping $\lambda_\text{spec} = \lambda_\text{div} = 5$. This is one of the strategies we followed during our evaluation and it yielded better results than most baselines without spending a significant amount of work fine-tuning our hyperparameters.
> >
> >
> > ### How does our evaluation show that TabCBM is more interpretable when we do not have ground-truth “meta-concepts”
> >
> > This is a great question. We first point out that quantitative evaluation of the alignment between discovered concepts/latent representations and what might be a concept to a domain expert in the absence of ground-truth labels is commonplace in the related field of disentanglement learning [2,3,4]. Such quantitative analysis, as opposed to pure qualitative approaches such as visual inspection, allows for easy benchmarking and evaluation of methods that are aiming to solve the same problem. Furthermore, it provides insights into how the representations learnt by a given method are able to capture the important components in a given task, leading to representations that may be interpreted in practice by experts via analysis like the ones described above. Inspired by this, in this work we deploy several well-known metrics in representation learning in order to obtain a quantitative evaluation of the concept explanations learnt by TabCBM with respect to explanations learnt by competing methods such as SENN and CCD. We believe that such evaluation is a crucial complement to any qualitative analysis (such as that done in Figure 4)  which, although interesting and insightful on its own, hides the variance that may exist in a model and forbids systematic comparison between existing state-of-the-art methods.
> >
> > ### Density and clarity of evaluation section
> >
> > Thank you for flagging a potential lack of clarity in our experimental section. To address this, we have made the following changes to our experimental section:
> > - We added a new Appendix (App. L) that shows a table containing a description of the disentanglement learning metrics used in Section 5.2.
> > - We explicitly clarified the main takeaway of each section and the experiments that will follow, at the beginning of each experimental section.
> >
> > Besides this, we would like to suggest that, if this is something other reviewers find agreeable, we can also move some of the metrics ($R^4$, Dis, and Compl) used in Table 2 to an Appendix to focus only on what we believe are the more significant metrics (CAS, MIG, and Info).

---

> > > ### Author Response · Authors · 2023-05-28
> > > **References**
> > >
> > > ### References
> > >
> > > [1] Kotliar, Dylan, et al. "Identifying gene expression programs of cell-type identity and cellular activity with single-cell RNA-Seq." Elife 8 (2019): e43803.
> > >
> > > [2] Eastwood, Cian, and Christopher KI Williams. "A framework for the quantitative evaluation of disentangled representations." International Conference on Learning Representations. 2018.
> > >
> > > [3] Locatello, Francesco, et al. "Challenging common assumptions in the unsupervised learning of disentangled representations." international conference on machine learning. PMLR, 2019.
> > >
> > > [4] Ross, Andrew, and Finale Doshi-Velez. "Benchmarks, algorithms, and metrics for hierarchical disentanglement." International Conference on Machine Learning. PMLR, 2021.